**Subject Category:**
Biology (whole organism)

ecology/environmental science/health and disease and epidemiology

climate change, livestock production, enteric fermentation, greenhouse gases, methane emissions, antibiotics

**Author for correspondence:**
Rebecca Danielsson
e-mail: rebecca.danielsson@slu.se

# Compound- and context-dependent effects of antibiotics on greenhouse gas emissions from livestock

Rebecca Danielsson[1], Jane Lucas[2], Josef Dahlberg[1], Mohammad Ramin[3], Sigrid Agenäs[1], Ali-Reza Bayat[4], Ilma Tapio[4], Tobin Hammer[5] and Tomas Roslin[6,7]

[1]Department of Animal Nutrition and Management, Swedish University of Agricultural Sciences, 75007 Uppsala, Sweden
[2]Department of Soil and Water Systems, University of Idaho, 875 Perimeter Dr, Moscow, ID 83844, USA
[3]Department of Agricultural Research for Northern Sweden, Swedish University of Agricultural Sciences, 901 83 Umeå, Sweden
[4]Production Systems, Natural Resources Institute Finland, 31600 Jokioinen, Finland
[5]Department of Ecology and Evolutionary Biology, University of Colorado at Boulder, Boulder, CO 80309, USA
[6]Department of Ecology, Swedish University of Agricultural Sciences, 75651 Uppsala, Sweden
[7]Department of Agricultural Sciences, University of Helsinki, FI-00014 University of Helsinki, Finland

RD, 0000-0001-7244-061X; JL, 0000-0002-3931-1864; TH, 0000-0002-7308-8440; TR, 0000-0002-2957-4791

The use of antibiotics in livestock production may trigger ecosystem disservices, including increased emissions of greenhouse gases. To evaluate this, we conducted two separate animal experiments, administering two widely used antibiotic compounds (benzylpenicillin and tetracycline) to dairy cows over a 4- or 5-day period locally and/or systemically. We then recorded enteric methane production, total gas production from dung decomposing under aerobic versus anaerobic conditions, prokaryotic community composition in rumen and dung, and accompanying changes in nutrient intake, rumen fermentation, and digestibility resulting from antibiotic administration. The focal antibiotics had no detectable effect on gas emissions from enteric fermentation or dung in aerobic conditions, while they decreased total gas production from anaerobic dung. Microbiome-level effects of benzylpenicillin proved markedly different from those previously recorded for tetracycline in dung, and did not differ by the mode of administration (local or systemic). Antibiotic effects on gas production differed substantially between dung maintained under aerobic versus anaerobic conditions and between

compounds. These findings demonstrate compound- and context-dependent impacts of antibiotics on methane emissions and underlying processes, and highlight the need for a global synthesis of data on agricultural antibiotic use before understanding their climatic impacts.

## 1. Introduction

Currently, antibiotics are widely used to treat and prevent infectious diseases of farm animals. In some countries, such as the USA, Canada, China and Russia, antibiotics are also used to promote growth and milk production [1,2]. Within the EU, it is not allowed to administer antibiotics for growth promotion [3]. Yet, there is significant variation in use between countries within the EU [4], which suggests that use often exceeds actual therapeutic needs. The use of antibiotics in farm animals may have severe consequences for the environment and for human health. Where the evolution of antibiotic resistance among pathogens is currently attracting much attention [5], the extensive use of antibiotics can have less-explored ramifications at the ecosystem level. In particular, antibiotic treatment may affect live microbial communities in two interconnected habitats: in the digestive tract of cattle, and in cattle manure. Each of these changes may have drastic effects on a major ecological disservice: the release of methane ($CH_4$) from cattle and their dung. Overall, it has been estimated that more than one-twentieth of all anthropogenic greenhouse gas emissions derive from cattle [6]. Given the enormous scale of these processes, the ecological ramifications of antibiotic use are potentially massive, but have not been sufficiently studied.

Ruminants have diverse microorganisms in their rumen [7,8], which are able to use substrates rich in cellulose. The fermentation mediated by microbes produces metabolic hydrogen ($H^+$). Yet, accumulation of $H^+$ is prevented by its immediate use by other microbes. A subset of the rumen microbiome consists of methanogenic archaea, which use hydrogen to reduce $CO_2$, thereby producing $CH_4$ and removing hydrogen [9]. The $CH_4$ produced is not used by the animal and is released by breath, burps and farts to the atmosphere, where it has a warming impact on the climate [10–12]. $CH_4$ production from ruminant livestock represents both a significant source of anthropogenic $CH_4$ emissions, and a loss of energy available to the cow [10–14]. Thus, over the last few decades, much research effort has been invested in finding strategies for lowering $CH_4$ emissions from ruminants [15–18]. Furthermore, research on the potential use of $CH_4$ from cattle manure for biogas production has also attracted much interest [19,20].

Against this background, our recent study [21] showed that cows treated with the broad-spectrum antibiotic tetracycline were characterized by changes in the microbial flora—resulting in a doubling of $CH_4$ emissions from dung. This suggested that the effect of antibiotics on gut microbes, which affect feed digestion within the cows, can have a significant impact on the global greenhouse effect. Since rumen microbial populations are the main degraders and fermenters of the feed, it is not surprising that feed digestion and $CH_4$ production might change when cattle are treated with antibiotics. At the same time, our study raised several new questions. First, where Hammer *et al.* [21] focused on antibiotic effects on $CH_4$ emissions from manure on pastures, $CH_4$ emissions from enteric fermentation are known to be significantly larger [6,15]. Thus, it is important to ask whether antibiotics have similar effects on $CH_4$ emissions from enteric fermentation. Second, different antibiotic compounds come with different modes of action (see the electronic supplementary material), targeting different microbes—and the same compound may be administered in multiple ways. How effects differ between compounds and uses is unclear. Finally, of cattle manure produced today, only a limited fraction will be deposited on pastures, where it will dry out and thus decompose under increasingly aerobic conditions [22]. In the temperate zone, which has a limited grazing season, this fraction is typically low [6], and a substantial—but poorly quantified—fraction will never reach the pasture, but end up in sludge and manure storage tanks. Of this, part is used for biogas production under anaerobic conditions. Thus, we ask how antibiotics affect gas emissions under aerobic field conditions versus anaerobic conditions akin to those prevailing in sludge.

With respect to these corollaries, Hammer *et al.* [21] hypothesized that (i) antibiotics in general increase $CH_4$ emissions, with the effects on emissions from manure being dwarfed by effects on enteric fermentation; (ii) that effects may vary among antibiotic compounds and (iii) with the mode of administration. In addition, it may be predicted (iv) that effects of antibiotics on gas production will differ between faeces decomposing in the field versus as sludge, given conditions favouring aerobic versus anaerobic organisms and processes. In the current paper, we revisit these hypotheses. In doing so, we shed new light on the overall effect of antibiotics on $CH_4$ emissions from cattle, and the mechanisms underlying these emissions.

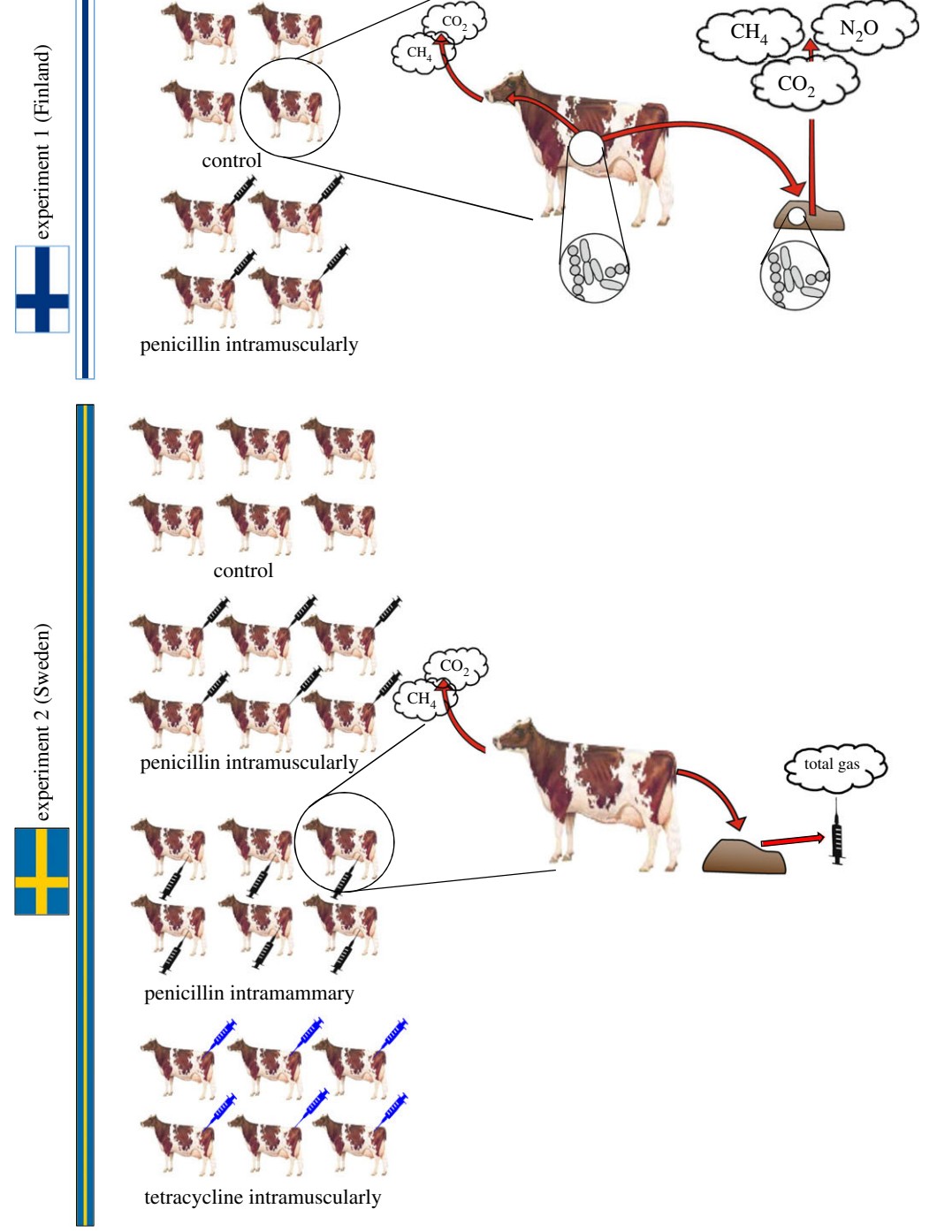

**Figure 1.** Study design. This study addresses the impacts of commonly used antibiotic compounds at multiple interconnected levels, including *in vivo* assays for their effects on microbial community structure, whole-animal methane production and fibre digestion; *in vivo* effects on microbial community composition and functional outcome in terms of feed digestion and methane production; *in situ* effects of antibiotic treatment on gas emissions from dung decomposing under field conditions, and *in vitro* effects on gas production from dung samples maintained under anaerobic conditions. As joint material, cows in two experimental herds were randomly selected for antibiotic treatment (syringes) and half used as untreated controls (no syringes).

## 2. Material and methods

To gain insight into each of the predictions, we targeted both the cattle rumen and cattle dung, testing for effects of two antibiotics on feed digestion and $CH_4$ emissions from lactating cows. We structured our approach into four steps (figure 1). First, to establish the effects of different antibiotic compounds on gas emissions at the whole-organism level, we measured gas emissions from dairy cows in two *in vivo*

experiments, referred to as experiment 1 and experiment 2. We used a randomized design to administer two different antibiotic compounds (benzylpenicillin or tetracycline) in two different ways (intramuscularly or intramammary; figure 1, left-hand side; note that the experiment was not fully factorial, and that the two experiments differed in terms of the compounds used and their modes of administration). We then measured gas emissions from enteric fermentation (see figure 1, mid-section) using metabolic chambers (experiment 1) or a GreenFeed system (experiment 2). Second, to establish the antibiotic treatment effects on gas emissions from cow pats decomposing under open-air conditions (see [21]), in experiment 1 we measured total $CH_4$, carbon dioxide ($CO_2$) and nitrous oxide ($N_2O$) emissions from four pats of each experimental cow over a four-week (28-day) time period (figure 1, top-right). Third, to establish the effects of antibiotic treatments on gas production under conditions akin to a biogas reactor (anaerobic processing of manure), in experiment 2 we measured total gas production of a standard amount of dung under anaerobic conditions maintained *in vitro* in closed syringes (figure 1, bottom-right). Finally, to investigate the functional basis for the patterns observed, in experiment 1 we studied changes in prokaryotic community composition in the rumen (figure 1, top row, blow-up of rumen) and dung (figure 1, top row, blow-up of dung), as well as nutrient intake, rumen fermentation and digestibility (i.e. utilization of feed). While effects of tetracycline on the microbiome were studied by Hammer *et al.* [21], we use the current study to probe for effects of another compound (procaine benzylpenicillin, henceforth 'penicillin' for brevity). For a complete list of processes monitored and metrics recorded, see table 1. For a specific timeline of treatment and sampling, see electronic supplementary material, figure S1.

## 2.1. Experimental design

For a graphical summary of the experimental design, see figure 1, for details on methods, animals as well as their care, see the electronic supplementary material.

### 2.1.1. Experiment 1 (Finland)

The effect of penicillin, the antibiotic most widely used in Nordic animal husbandry, was tested in Natural Resources Institute Finland (Luke). Methane emissions were measured using respiration chambers. Daily feed intake, milk yield and manure excretion were measured simultaneously.

Due to the restricted number of respiration chambers ($N = 4$), a maximum of eight cows could be included in the study. Those eight cows were selected from the larger herd of more than one hundred cows held in the experimental barn, following standardized criteria: the recruited cows had not been previously treated with antibiotics and were all in mid lactation. The selected cows were split in pairs, and within each pair, the antibiotic treatment was assigned randomly. To match the availability of respiration chambers, cows were split into two groups, with two treatment and two control cows in each. These groups of animals were fed and sampled following identical methodology, with an interval of one week. They are henceforth referred to as 'group 1' and 'group 2' (for a comprehensive timeline of the experiment, see electronic supplementary material, figure S1).

Cows were offered the routine diet used in the research barn (see the electronic supplementary material) and diet composition remained constant during the whole experimental period. The study started with a 10-day control period (henceforth referred to as the 'baseline period'), during which the cows were first kept in the respiration chambers for 4 days to establish baseline levels of gas production (see below), and then released to the barn for a further 6 days. After the baseline period cows were treated with procaine benzylpenicillin, Penovet® vet. (suspension 300 mg ml$^{-1}$; Boehringer Ingelheim Vetmedica) 20 000 IU kg$^{-1}$ injected intramuscularly once per day for 5 consecutive days. During the last day of treatment, cows were moved to respiration chambers (henceforth 'post-treatment period 1'), with a third session in the chambers occurring two weeks after the last day of treatments (henceforth 'post-treatment period 2'). Each measurement period in the chambers lasted for 4 days (1 day adaptation and 3 days measurement).

### 2.1.2. Experiment 2 (Sweden)

To establish the effect of two antibiotic compounds administered in different ways, we performed a second experiment in the research herd of the Swedish University of Agricultural Sciences in Umeå, Sweden. The study involved 24 healthy lactating Nordic Red dairy cows. In total the study lasted eight weeks, during which mass emission of $CH_4$ and $CO_2$ was measured by a head chamber system

**Table 1.** A complete list of processes measured, including the metrics derived, the techniques used in their measurement and a few notes on their meaning.

| process and metrics | unit measured in | method used to quantify |
|---|---|---|
| *nutrient intake* | | |
| dry matter (DM) intake[a] | kg d$^{-1}$ | feed consumed × feed dry matter content measured in oven (60°C, 48 h) |
| organic matter (OM) intake[b] | kg d$^{-1}$ | subtracting ash (510°C for 16 h) from DM intake[c] |
| crude protein (CP) intake[d] | kg d$^{-1}$ | nitrogen measured by Dumas method multiplied by 6.25[e] |
| neutral detergent fibre (NDF) intake[f] | kg d$^{-1}$ | neutral detergent solution[g] |
| *rumen fermentation*[h] | | |
| hydrogen ion pressure[i] | 0–14 | portable pH meter[j] |
| ammonia concentration[k] | mmol l$^{-1}$ | direct colorimetric method[l] |
| volatile fatty acids (VFA)[m] | mmol l$^{-1}$ | gas chromatography[n] |
| acetic acid[o] | mmol mol$^{-1}$ VFA | gas chromatography[n] |
| propionic acid[p] | mmol mol$^{-1}$ VFA | gas chromatography[n] |
| butyric acid[q] | mmol mol$^{-1}$ VFA | gas chromatography[n] |
| isobutyric acid[r] | mmol mol$^{-1}$ VFA | gas chromatography[n] |
| valeric acid[s] | mmol mol$^{-1}$ VFA | gas chromatography[n] |
| isovaleric acid[t] | mmol mol$^{-1}$ VFA | gas chromatography[n] |
| caproic acid[u] | mmol mol$^{-1}$ VFA | gas chromatography[n] |
| molar ratio of acetate to propionate[v] | n.a. | derived from metrics above[n] |
| *digestibility (i.e. utilization of feed)*[w,x] | | |
| digestibility of OM | g kg$^{-1}$ | indigestible NDF as internal marker; total collection of faeces |
| digestibility of CP | g kg$^{-1}$ | indigestible NDF as internal marker; total collection of faeces |
| digestibility of NDF[y] | g kg$^{-1}$ | indigestible NDF as internal marker; total collection of faeces |
| digestibility of potentially digestible NDF[z] | g kg$^{-1}$ | indigestible NDF as internal marker; total collection of faeces |
| *milk production*[aa] | | |
| milk production | kg d$^{-1}$ | direct measurement |
| *gas emissions from enteric fermentation* | | |
| total methane (CH$_4$)[ab] | g d$^{-1}$ | respiration chambers, Columbus Instruments, OH, USA. GreenFeed C-Lock Inc., Rapid City, South Dakota, USA. |
| methane yield[ac] | g kg$^{-1}$ DM intake | derived from metrics above |
| methane emissions/unit OM intake[ad] | g kg$^{-1}$ OM intake | derived from metrics above |
| methane intensity[ae] | g kg$^{-1}$ milk | derived from metrics above |
| total carbon dioxide (CO$_2$) | g d$^{-1}$ | respiration chambers, Columbus Instruments, OH, USA |
| CO$_2$ yield[af] | g kg$^{-1}$ DM intake | derived from metrics above |
| CO$_2$ emissions/unit OM intake | g kg$^{-1}$ OM intake | derived from metrics above |
| CO$_2$ emissions/unit NDF intake | g kg$^{-1}$ NDF intake | derived from metrics above |
| CO$_2$ intensity | g kg$^{-1}$ milk | derived from metrics above |
| *gas emissions from dung under pasture conditions* | | |
| total methane (CH$_4$) | g m$^{-2}$ h$^{-1}$ | portable FTIR instrument[ag] |
| total carbon dioxide (CO$_2$) | g m$^{-2}$ h$^{-1}$ | portable FTIR instrument[ag] |
| total nitrous oxide (N$_2$O) | g m$^{-2}$ h$^{-1}$ | portable FTIR instrument[ag] |

(*Continued.*)

**Table 1.** (*Continued.*)

| process and metrics | unit measured in | method used to quantify |
| --- | --- | --- |
| *gas production from faeces in vitro* | | |
| total gas production | ml gas/g DM/week | gas accumulation in in airtight glass syringes[ah] |
| *prokaryotic community composition* | | |
| no. and identity of prokaryote sequences | DNA sequence data | metabarcoding of 16S rRNA[ai] |

[a]Critical measure linked to the productivity of animal and can be affected by many factors including body weight, physiological stage, production level, diet quality, dietary treatments etc.

[b]Highly correlated with DM intake as ash content of diet is rather constant. Represents the fraction of feed which nutrients rather than minerals are extracted from.

[c]AOAC-942.05.

[d]Representing sum of true protein and non-protein nitrogen.

[e]AOAC-968.06.

[f]Representing the structural carbohydrates of feed (fibre).

[g][23].

[h]VFA and ammonia-N determinations by procedures described in [24]. Samples of rumen fluid were stored at −20℃ until analysis.

[i]Linked with rumen fermentation and health. Low rumen pH causes acidosis.

[j]Measured immediately upon sampling with a Mettler-Toledo AG unit, Schwerzenbach, Switzerland.

[k]Indicator of N degradation, N utilization by microbes and its absorption from rumen wall.

[l][25].

[m]An indicator of feed fermentation in the rumen. Can be affected by many factors including dietary treatments.

[n][26].

[o]Higher in high forage diets and lower in high concentrate diets.

[p]Lower in high forage diets and higher in high concentrate diets.

[q]Increases with diets containing high sugar content.

[r]A branched chain VFA which is related to degradation of amino acids.

[s]A minor VFA.

[t]A branched chain VFA which is related to degradation of amino acids.

[u]A minor VFA.

[v]An indicator of ruminal fermentation type. This is higher with high forage diets and lower with high concentrate diets.

[w]The proportion of feed which is not excreted in faeces. It can be affected by different factors including feed quality or dietary treatments.

[x]Total-tract apparent digestibility coefficients determined using either indigestible neutral detergent fibre (NDF) as an internal marker (baseline period and post-treatment period 1) or total faecal collection (post-treatment period 2).

[y]NDF is an index of fibre content 'neutral detergent fibre'.

[z]pdNDF represents the proportion of fibre (NDF) that is potentially digestible, determined by incubation in the rumen to measure indigestible NDF, then subtracting it from NDF to obtain pdNDF.

[aa]Affected by many factors (similar to DM intake) and dietary treatments.

[ab]A by-product of feed fermentation in rumen. About 95% produced in the rumen and 5% in the hindgut. Can be affected by dietary and microbial factors. Related mainly to the intake level and fibre intake of animals.

[ac]Decreases when the production level increases. This parameter corrects for variation in the feed intake. It reduces slightly with increasing intake level or better utilization of feed.

[ad]Decreases when the production level increases. This parameter corrects for variation in the OM intake. It reduces slightly with increasing intake level or better utilization of feed.

[ae]Indicates methane produced per unit of milk. Declines with increasing milk yield or better utilization of feed.

[af]The sum of enteric and respiratory $CO_2$ which is affected by both rumen fermentation and animal metabolism.

[ag]Gasmet™ DX4015; Gasmet technologies, Espoo, Finland.

[ah][27,28].

[ai]Following Lucas *et al.* [26].

(GreenFeed system, C-Lock Inc., Rapid City, South Dakota, USA; [29]), with details given in the electronic supplementary material.

For the experiment, the cows were blocked according to milk somatic cell count parity (i.e. the individual rank number of the current lactation period) and stage of lactation, then allocated to three treatment groups and one control group, with six cows per group. This procedure created treatment and control groups of identical composition, featuring the inter-individual variation typical of any

cow herd. As treatment, cows were administered (i) a 5-day treatment of procaine benzylpenicillin, Penovet® vet. (Boehringer Ingelheim Vetmedical), with 20 mg kg⁻¹ BW intramuscularly one time per day (henceforth 'penicillin intramuscularly'); (ii) a 5-day intramammary treatment of procaine benzylpenicillin, Carepen® vet. (Boehringer Ingelheim Vetmedical) 600 mg injected into the right front udder quarter (henceforth 'penicillin intramammary'), or (iii) a 4-day treatment of oxytetracycline, Engemycin® vet. (Intervet) 10 mg kg⁻¹ BW given intramuscularly once per day (henceforth 'tetracycline intramuscularly'). All treatments were performed according to the instructions provided by the manufacturers. For further justification for the compounds selected and the doses used, see the electronic supplementary material.

The study started with a 14-day adaptation period to the diet, housing and GreenFeed system [29] and then continued with a control week with the first sampling occasion (henceforth referred to as the 'baseline period'). This period was followed by the 'treatment period' during which the antibiotics were administered, followed by a period of four weeks during which responses were recorded. While CH₄ emissions and feed intake were continuously recorded during the full experiment, other sampling (including measurement of milk production) was focused on two time periods: a first period of 5 days following immediately after the last day of antibiotic administration, and a second period following 30 days after the end of antibiotic treatment and hence 25 days after the first sampling period. For consistency with experiment 1, we henceforth focus our analyses on these discrete time periods, referring to them as post-treatment period 1 and 2. Relative to the end of antibiotic treatment, post-treatment period 1 coincided across the two experiments, but post-treatment period 2 came 11 days later in experiment 2 than in experiment 1 (electronic supplementary material, figure S1). This was an explicit decision allowing for longer recovery time in experiment 2 than in experiment 1, the end of which was constrained by the start of the compulsory outdoor grazing season for Finnish dairy cattle. For a complete timeline, and for details on the feed and care of animals during the experiments, see the electronic supplementary material.

## 2.2. Quantification of gas production *in vivo*

To quantify the effects of antibiotics on enteric CH₄ production from the cows, we used two standardized methods as available in the two different barns. In experiment 1, CH₄ and CO₂ production was measured in hermetically sealed respiration chambers, allowing the accurate recording of CH₄ and CO₂ in and out of the chamber. In experiment 2, CH₄ and CO₂ production was measured by an open-circuit head chamber (GreenFeed system, C-Lock Inc., Rapid City, South Dakota, USA; [29]). The cows visited the head chamber several times per day to get pelleted concentrates. During each visit, the production of CH₄ and CO₂ was measured, with details of the feed offered in the electronic supplementary material.

### 2.2.1. Experiment 1 (Finland)

Methane emissions were measured using respiration chambers (Oxymax, Columbus Instruments, OH, USA) located in the Luke research barn. An adjustable air conditioning system (flow 500–2000 l min⁻¹; cooling capacity 2.9 kW; heating capacity 3.2 kW) allows environmental control of temperature across a range of 15–22°C and a relative humidity of no more than 70%. Temperature and relative humidity inside the chambers were monitored using electronic sensors (Vaisala OY, Finland). Concentrations of CO₂, O₂ and CH₄ in the exhaust air flow were measured using dedicated gas analysers (Columbus Instruments, OH, USA). Gas analyses were configured to allow automatic measurements at 3.5 min intervals from each chamber and the reference air. Prior to every test period, automatic zero and span calibration of analysers was conducted using standard gases (AGA, Ltd, Espoo, Finland). Signal outputs from monitoring equipment and gas analysers were connected to a data logger allowing for automatic data recording (Oxymax, Columbus Instruments, OH, USA). Each chamber period lasted for 4 days (1 day for adaptation and 3 days for measurements; for a complete timeline, see electronic supplementary material, figure S1). For a complete list of metrics derived, technique for measurement and meaning, see table 1.

While in the chambers, the cows will naturally defecate, raising the question of how much gas emissions from dung may contribute to the overall fluxes recorded. However, all dung was removed from the chambers twice a day—a practice which drastically reduced the contribution of minor emissions from dung.

### 2.2.2. Experiment 2 (Sweden)

Mass emission of $CH_4$ and $CO_2$ in exhaust gases was measured by a portable open-circuit head chamber system (GreenFeed system) as described by Huhtanen *et al.* [29]. The animals receive a small amount of concentrate at each visit in the GreenFeed system. In the present study, the system was programmed to allow each animal a visit every 5 h. During each visit, cows were given eight drops of 50 g of concentrate at 40 s intervals. Airflow rates and gas concentrations are measured continuously in the system, and by using the gas sensor information, the volumetric flux ($l\,min^{-1}$) of gases emitted by the animal can be calculated.

As was the case in experiment 1 (see above), gas emissions from dung will not affect the overall fluxes recorded in experiment 2. The GreenFeed system is explicitly designed to record enteric methane production from breath and burps. For this purpose, it uses a non-dispersive infrared (NDIR) sensor to monitor gas contents in the air space directly in front of the cow's mouth. Since the system directly records head position during the visit, we were able to filter out any data recorded during inadequate head position. For a note on the high reliability of the GreenFeed system, see the electronic supplementary material.

## 2.3. Quantification of antibiotic effects on feed digestion

To evaluate the effects of antibiotics on feed digestion, we first recorded nutrient intake (see above and table 1). To score digestibility, we collected total faeces over 3 days during post-treatment period 2 in experiment 1 (Finland; electronic supplementary material, figure S1), and submitted representative samples for chemical analysis. Total-tract apparent digestibility coefficients were determined by using either indigestible neutral detergent fibre (NDF) as an internal marker (baseline period and post-treatment period 1) or total faecal collection over a 72 h interval starting at 18.00 on day 36 of post-treatment period 2 (electronic supplementary material, figure S1). Faeces were weighed, thoroughly mixed, subsampled (5%, wt/wt) and stored at −20°C before chemical analysis. To determine the indigestible fraction of NDF (baseline period and post-treatment period 1) in silage, concentrate and faeces, we incubated duplicate 1.0 g samples of dry matter (DM) in nylon bags (60 × 120 mm, pore size 0.017 mm) within the rumen of two cows fed a grass silage-based diet (forage to concentrate ratio = 70 : 30 on a DM basis) for 288 h. Once removed from the rumen, the bags were rinsed in cold water for 25 min using a household washing machine, incubated for 1 h in boiling neutral detergent solution, rinsed and dried to a constant weight at 60°C. Total-tract apparent digestibility of nutrients was then calculated as 100 − (%marker in feed/%marker in excreta) × (%nutrient in excreta/%nutrient in feed) × 100 or ((nutrient intake − nutrient excretion)/nutrient intake) × 100 for marker and total collection methods, respectively.

## 2.4. Characterization of the prokaryotic community composition of rumen and dung

To analyse the antibiotic effect on the ruminal microbial composition in experiment 1, rumen fluid was collected twice from each animal by a vacuum hose supplied via the mouth [30]. This sampling was repeated during the baseline period, post-treatment period 1 and post-treatment 2, in each case just before the cows entered the chambers and right after they left the chambers (see timeline in electronic supplementary material; electronic supplementary material, figure S1).

To describe the microbiome of dung before and after antibiotic treatment of experiment 1 (Finland), 0.5 g of faecal grab samples were collected in duplicate from each cow at Luke one week prior to antibiotic treatment, right after and two weeks after treatment (electronic supplementary material, figure S1).

Rumen fluid and faecal samples were immediately preserved on dry ice and transferred to −80°C until further analyses. After quick thawing, samples were placed in Zymo Xpedition lysis stabilizing solution and were disrupted for 30 s using a TissueLyser II (Qiagen, Hilden, Germany). After tissue disruption, all samples were placed in a −80°C freezer until subsequent DNA extractions could be performed. DNA was extracted using Zymo Xpedition Soil/Fecal sampling kit (Qiagen, Hilden, Germany) following the manufacturer's protocols.

Once DNA was extracted, we amplified the V4 hypervariable region of the 16S rRNA gene using 515F 5′-GTGCCAGCMGCCGCGGTAA-3′ and 806R 5′-GGACTACHVGGGTWTCTAAT-3′ primers [31]. The 515F primer was modified to include a 16 bp M13 sequence (GTAAAACGACGGCCAG) at the 5′ end to allow for the attachment of a unique 12 bp barcode in a subsequent PCR reaction. Barcoded

amplicons were cleaned and product was standardized using SequalPrep Normalization plates (ThermoFisher, Inc.; [32]). Amplicons were sequenced using a MiSeq instrument with 500 V2 chemistry for paired end reads ($2 \times 250$ bp).

Illumina sequencing reads were analysed and demultiplexed using QIIME v. 1.9.1 [33]. Sequencing reads that contained errors in the barcoded region, ambiguities, homopolymers (greater than six nucleotides in length) or an average quality score less than 25 were discarded. Primer sequences were trimmed, and chimeric sequences were eliminated using USEARCH (version 6.1) and the 'gold' reference database for bacterial and archaea [34]. All sequences were clustered into de novo operational taxonomic units (OTUs) at 97% similarity. Bacterial and archaeal taxonomic classification was assigned via the SILVA reference database (release 119; [35]) using the pyNAST alignment algorithm. Prior to all statistical analyses, we rarefied (randomly subsampled) samples to an even sequencing depth of 2170 reads/sample for rumen analysis and 1000 reads/sample for dung analysis.

## 2.5. Quantification of gas emissions from dung under field conditions

To establish the effects of antibiotic treatments on gas emissions from cow pats decomposing under open-air conditions (see [21]), we measured total $CH_4$, $CO_2$ and $N_2O$ emissions from four pats of each cow. To initiate the experiment, we collected 6–9 kg of dung rectally from each cow into a separate, clean tub with a volume of 10 l. Due to the antibiotic treatment schedule (electronic supplementary material, figure S1), the cows were split into two groups with two treatment and two control cows per group. Group 1 was sampled on 30 April 2017, and group 2 was sampled on 6 May 2017, in each case within a single day (between 09.30 and 16.00, beginning 1 h after the last administration of antibiotics). For the remainder of the experiment, pats from group 1 and group 2 were sampled with the same methodology, one week apart.

On the day of dung collection, the dung from each cow was homogenized and split into nine pats of 0.75 l each, using a scoop replaced between cows. Pats were placed into separate pie tins, and immediately transferred to a semi-natural field located at the Viikki Experimental farm in Helsinki. Here, the land had been used for grazing cattle six months prior to our experiment. Our pasture land was fenced off to keep any nearby grazing cattle away from our pats. Similarly, all pats were placed at least 2 m away from the fence line to decrease the potential of leaching from non-experimental pats. Pats were transferred from their pie tins onto 4 $cm^2$ mesh screens and placed in direct contact with the pasture grass. Mesh covers 25 cm × 25 cm × 8 cm in dimension were placed over pats to keep avian pests from disrupting pats and consuming colonizing invertebrates. Mesh size of the covers was 4 $cm^2$, which allows for most colonizing invertebrates to gain access to pats.

$CO_2$, $CH_4$ and $N_2O$ production from the pats were measured in the field using a portable continuous gas analyser GASMET DX-4030 (Gasmet Technologies Oy, Helsinki, Finland). Pats were measured multiple times throughout the experiment (on days 1, 4, 8, 14 and 28 after establishment). During measurements, a chamber was placed over each pat with a circulation fan inside to sustain airflow. These chambers push 2.5 cm into the soil to ensure a tight seal over pats, preventing external gas flow from being measured. Pats were measured continuously for 5 min. Temperature within the chamber was taken at the beginning of each measurement. To get background levels of gas emissions, three measurements for each time point were taken of the soil surrounding the focal pat.

Weather during the spring was cold, with the first week of group 1 including sub-zero night-time temperatures (electronic supplementary material, figure S2). For this reason, the effect of group identity and its interaction with other factors was included in all statistical models of the resultant gas emissions.

## 2.6. Quantification of gas production from faeces *in vitro*

To establish the effects of antibiotic treatments on gas production from dung under anaerobic conditions, we measured the total gas production from faecal samples collected in experiment 2. Approximately 40 mg of faecal sample was collected from the rectum from each of the 24 cows on the last day of antibiotics treatment. These samples were then aliquoted in glass syringes [27], with the piston of the syringes used to press out any extra gas. A suba-seal rubber septum (Merck, Darmstadt, Germany) was fitted to the end of the syringe and the baseline value ($V_0$) of the syringe content was recorded. Syringes were kept at room temperature (20°C) throughout incubation period, with gas production recorded once a week for the first eight weeks, and thereafter every second week. If the gas production exceeded 80 ml on the syringe, gas was pressed out and the piston was set to $V_0$ again.

Between the last two weeks of readings no gas was produced and the measurements were stopped at day 96 after the last day of antibiotics treatment. At an early stage of incubation, two syringes in the 'penicillin intramammary' treatment, and three syringes in each of the 'tetracycline intramuscularly' and 'penicillin intramuscularly' treatment started to mould, and were hence excluded from the dataset.

## 2.7. Statistical analyses

Data on all responses (barring exceptions below) were fitted to the following response-specific, univariate statistical models, using the MIXED procedure of SAS (9.3, SAS Inst., Inc., Cary, NC). In all our analyses, the main interest was in the main effect of treatment (identifying whether mean responses differed more between than within treatments) and in interactions involving treatment.

$$Y = \mu + G + T + P + T \times P + \text{covariate} + \varepsilon,$$

where $Y$ is the dependent variable, $\mu$ is the mean for all observations; $G$ is the effect of group (see above), $T$ is the effect of treatment; $P$ is the effect of period (post-treatment period 1 versus 2) and $T \times P$ (identical to $T(P)$) is the effect of treatment within period. To account for individual variation, emissions measured for each cow during the control period were used as a covariate in the model (covariate), and $\varepsilon$ is the random residual error. To account for cow-to-cow variation in repeated measures of the same animal, we used cow identity as a categorical random effect. Here, the term $T \times P$ quantifies whether the recovery rate from a potential response to treatment varied with the treatment in question.

Data on gas emissions from faeces under field conditions in experiment 1 (Finland; see *Quantification of gas emissions from dung under field conditions*) were fitted to the following, compound-specific univariate models:

$$Y = \mu + T + G + D + T \times D + T \times G + G \times D + T \times D \times G + \varepsilon$$

using the same notation as above. Here, terms $T \times G$ and $T \times D \times G$ capture variation in temporal emission patterns between groups 1 and 2 as due to differences in weather conditions (see the electronic supplementary material for a description of weather events). To account for autocorrelation among repeated samples taken within the same cow and dung pat, we assumed a temporal power covariance structure, with the sample as the repeated subject and day as the coordinate for distance between observations.

Data on gas production from faeces, in experiment 2 (Sweden), under *in vitro* conditions were fitted to the following model using the MIXED procedure:

$$Y = \mu + T + W + T \times W + \varepsilon$$

using the same notation as above, but introducing W for the effect of week in the syringe, and $T \times W$ for the effect of treatment within week (identical to $T(W)$). We accounted for autocorrelation among repeated samples taken from the same unit by assuming a temporal power covariance structure, with the sample as the repeated subject and week as the coordinate for distance between observations.

For all models, we used a Kenward–Roger approximation to derive relevant degrees of freedoms. Least square means were calculated using the LSMEANS/Slice = time and where appropriate (i.e. for significant type 3 F-tests), the statistical significance of differences between treatments were determined following a Tukey adjustment. Differences were treated as significant at $p < 0.05$.

### 2.7.1. Prokaryotic community structure

As a measure of differences in bacterial community composition among samples (experiment 1), we used weighted UniFrac distances [36]. To visualize impacts of time and treatment on bacterial composition, we created nonmetric multidimensional scaling (NMDS) ordinations and plotted them using the vegan package in R [37]. To test for differences in microbial community structure between treatments and between individual cows, we used permutational MANOVA, as implemented in QIIME v. 1.9.1 [33] with 999 permutations.

## 3. Results

We found strong effects on gas production from dung maintained under anaerobic conditions *in vitro*. These effects were reflected in antibiotic-induced shifts in the prokaryotic community composition of cow dung,

**Table 2.** Results from GLMM models of gas emissions from cows under *in vivo* conditions, measured in total grams per day. Shown are type 3 tests of fixed effects on (A) $CH_4$ emissions in experiment 1 (Finland) and (B) $CH_4$ emissions in experiment 2 (Sweden). Baseline refers to pre-treatment emissions, included as a covariate, whereas group in experiment 1 refers to two experimental cohorts of cows entering the metabolism chambers at different dates. Period refers to time since treatment (a discrete factor; see electronic supplementary material, figure S1 for a comprehensive timeline). Note that the baseline effect is always substantial, suggesting that cows characterized by initially high emissions remain so throughout the experiment. For results on additional responses, see the electronic supplementary material. Italics indicate *p*-value < 0.05.

| effect | NDF | DDF | F | p |
|---|---|---|---|---|
| (A) | | | | |
| baseline | 1 | 4 | 38.39 | *0.0034* |
| group | 1 | 4 | 3.63 | 0.1296 |
| period | 1 | 6 | 27.15 | *0.0020* |
| treatment | 1 | 4 | 0.03 | 0.8738 |
| treatment × period | 1 | 6 | 2.53 | 0.1625 |
| (B) | | | | |
| baseline | 1 | 17 | 44.20 | *<0.0001* |
| period | 1 | 18 | 14.66 | *0.0012* |
| treatment | 3 | 17 | 1.14 | 0.3619 |
| treatment × period | 3 | 18 | 0.31 | 0.8195 |

but not in the rumen microbiome. However, our experiment revealed no statistically detectable effect of antibiotic treatment on gas emissions from enteric fermentation, feed consumption or milk production from animals *in vivo*, or gas emissions from dung maintained under field conditions *in situ*.

## 3.1. Gas emissions from enteric fermentation

In both experiment 1 and experiment 2, gas emissions varied substantially between cows during the pre-treatment baseline period (electronic supplementary material, figure S6). Cows emitting high levels of $CH_4$ before the experiment maintained high levels throughout the experiment (table 2). Emissions also varied between time periods (table 2), but antibiotic treatment did not modify temporal patterns in either experiment 1 or experiment 2 (table 2; figure 2). In other words, none of the antibiotic treatments detectably modified gas emissions from enteric fermentation. Since antibiotic treatment did not detectably affect milk production or nutrient intake (see below and electronic supplementary material, tables S2, S5 and S6), measures of $CH_4$ yield scaled by intake were also unaffected by antibiotic treatment (electronic supplementary material, table S6).

## 3.2. Gas emissions from dung under field conditions

Methane emissions from dung pats exposed to field conditions changed strongly over time (table 3: main effect of day; figure 3), with differences in absolute emissions among the two groups of pats started one week apart (table 3). Emissions also changed at different rates in the two groups started at different dates (table 3), probably due to differences in weather conditions during establishment and hence in desiccation rates (compare figure 3 versus electronic supplementary material, figure S2). For $CH_4$ and $N_2O$, no significant differences related to treatment were detected (table 3).

## 3.3. Gas production from dung *in vitro*

Faecal gas production varied with treatment (table 4), and was inhibited by all three antibiotic treatments compared to the control (figure 4; $p < 0.001$). However, emissions from cows treated with penicillin injected intramuscularly versus intramammary did not detectably differ (figure 4; $p < 0.05$ for all pairwise comparisons). Overall, the effect sizes were large: over the 96-day measuring period, the control samples emitted a total of 13.9 (s.e. ± 0.61) ml gas/g DM, while faeces from cows receiving intramammary penicillin released 11.4 (s.e. ± 0.75) ml gas/g DM, faeces from cows receiving intramuscular penicillin

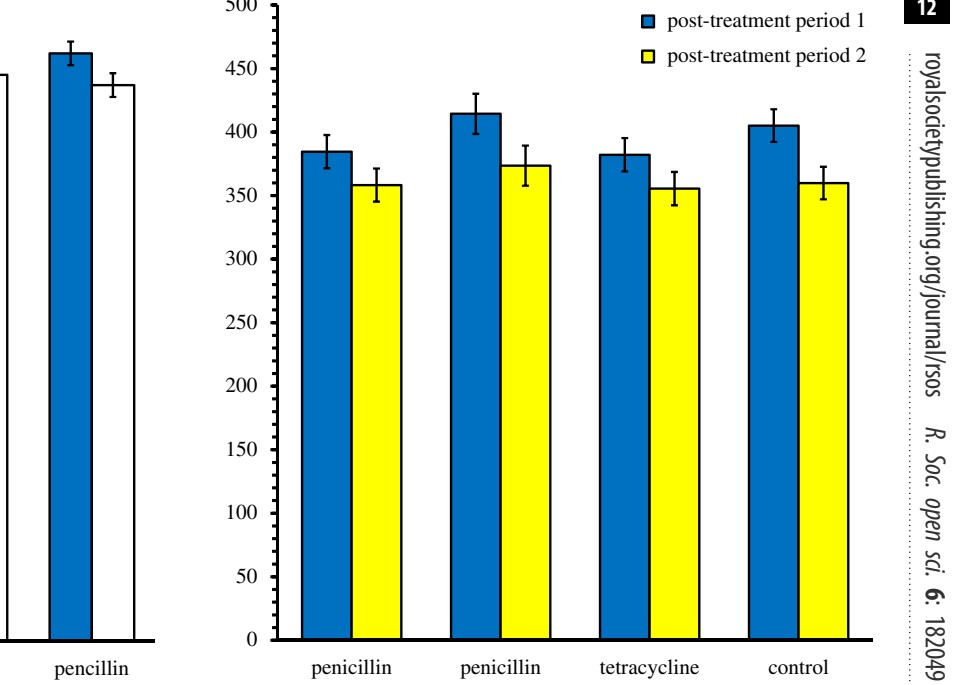

**Figure 2.** Methane emissions from enteric fermentation of whole cows *in vivo*, as observed in experiments 1 and 2. The figure shows least-squares means (±s.e.) from the GLMM described in table 2. Blue bars refer to measurements taken a few days after antibiotic treatment (period 1), light bars (yellow and white) to measurements taken four weeks after treatment (period 2; for a complete timeline of the respective experiments, see the electronic supplementary material). To separate between results from experiment 1 (Finland) and experiment 2 (Sweden), bars are identified by the national colours (blue and yellow for Sweden, blue and white for Finland). Note similarities in absolute emissions across the two experiments, despite some differences in methodology.

emitted 10.6 (s.e. ± 0.86) ml gas/g DM and faeces from cows receiving intramuscular tetracycline emitted 6.62 (s.e. ± 0.86) ml gas/g DM. Thus, treatment with tetracycline effectively halved total gas production.

## 3.4. Feed intake, rumen fermentation and digestibility

Antibiotic treatment had no consistent, detectable effect on metrics of feed intake, rumen fermentation or digestibility (see electronic supplementary material, tables S2 and S3 and figures S3 and S4 for specific results).

## 3.5. Prokaryotic community composition

Antibiotic treatment with penicillin, in experiment 1, had no detectable effect on the prokaryotic community composition of the rumen, but did impact the community composition of cow dung.

### 3.5.1. Rumen microbial communities

Individual cows differed substantially from each other in terms of their microbial community profile (Permanova pseudo-$F_{7,39}$: 1.56, $p = 0.001$). While cows in different treatments (control versus penicillin treatment) fell in somewhat different positions within the NMDS plots (figure 5), the same grouping was evident both before and after penicillin treatment (figure 5). The treatment effect was not significant in any of the periods, whereas the effect of cow individual was significant in all cases. Cow microbiomes differed with time, regardless if they were part of the control group or treated with penicillin (Permanova pseudo-$F_{3,44}$: 1.49, $p = 0.016$), whereas we found no change in the rumen microbiome between control and treatment groups during the baseline period (Permanova pseudo-$F_{1,13}$: 1.02, $p = 0.48$), post-treatment period 1 (Permanova pseudo-$F_{1,13}$: 1.04, $p = 0.37$) or post-treatment period 2 (Permanova pseudo-$F_{1,13}$: 0.91, $p = 0.62$). We did observe distinct microbiomes between our

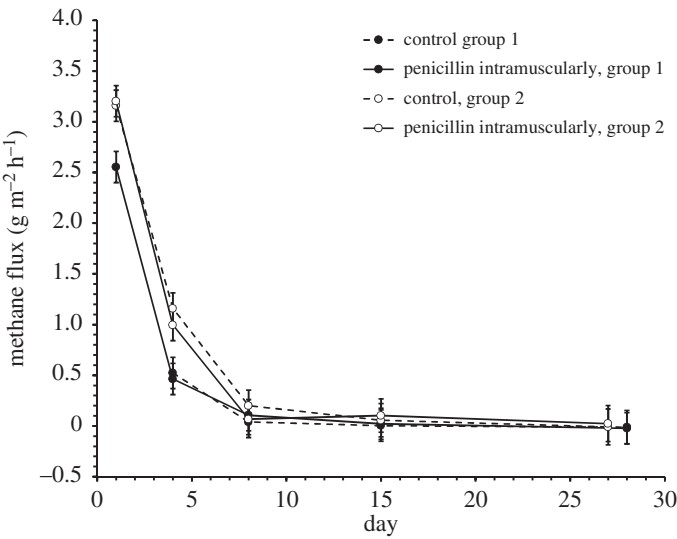

**Figure 3.** Methane emissions from cow pats exposed under field conditions. The figure shows least-squares means (±s.e.) from the GLMM described in table 3(A).

**Table 3.** Results from a GLMM model of gas emissions from manure under field conditions: (A) $CH_4$, (B) $N_2O$ and (C) $CO_2$. Shown are type 3 tests of fixed effects. Group refers to two experimental cohorts of cows entering the metabolism chambers at different dates (a discrete factor with levels 30 April 2017, and 6 May 2017, respectively; see electronic supplementary material, figure S1 for a comprehensive timeline). Italics indicate $p$-value < 0.05.

| effect | NDF | DDF | F | p |
|---|---|---|---|---|
| (A) | | | | |
| treatment | 1 | 28 | 2.74 | 0.1089 |
| group | 1 | 28 | 5.76 | *0.0233* |
| day | 5 | 104 | 234.62 | *<0.0001* |
| treatment × day | 5 | 104 | 2.06 | 0.0764 |
| group × day | 3 | 104 | 2.94 | *0.0366* |
| treatment × group | 1 | 28 | 2.33 | 0.1382 |
| group × treatment × day | 3 | 104 | 4.56 | *0.0048* |
| (B) | | | | |
| treatment | 1 | 28 | 0.14 | 0.7097 |
| group | 1 | 28 | 8.28 | *0.0076* |
| day | 5 | 104 | 3.30 | *0.0084* |
| treatment × day | 5 | 104 | 0.83 | 0.5287 |
| group × day | 3 | 104 | 0.99 | 0.3984 |
| treatment × group | 1 | 28 | 1.64 | 0.2109 |
| group × treatment × day | 3 | 104 | 0.82 | 0.4832 |
| (C) | | | | |
| treatment | 1 | 28 | 0.08 | 0.7777 |
| group | 1 | 28 | 42.61 | *<0.0001* |
| day | 5 | 104 | 14.17 | *<0.0001* |
| treatment × day | 5 | 104 | 1.05 | 0.3941 |
| group × day | 3 | 104 | 7.56 | *0.0001* |
| treatment × group | 1 | 28 | 3.09 | 0.0898 |
| group × treatment × day | 3 | 104 | 0.97 | 0.4089 |

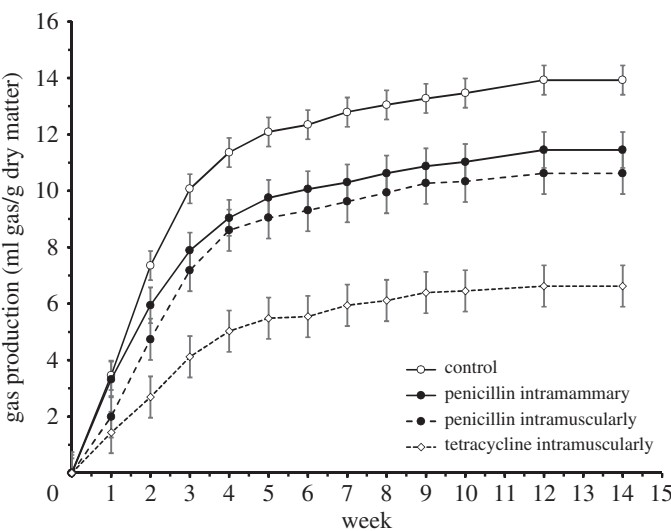

**Figure 4.** Gas production from faeces (ml gas/g dry matter since previous measurement) under anaerobic conditions *in vitro*. The figure shows least-squares means (±s.e.) from the GLMM described in table 4. Note that the standard errors (and hence confidence limits) of penicillin administered intramuscularly versus intramammary overlap at each time point, with no pairwise significant differences detected.

**Table 4.** Results from a GLMM model of faecal gas production under *in vitro* conditions. Shown are type 3 tests of fixed effects.

| effect | NDF | DDF | F | p |
|---|---|---|---|---|
| treatment | 3 | 155 | 20.06 | <0.0001 |
| week | 12 | 155 | 116.47 | <0.0001 |
| treatment × week | 36 | 155 | 3.64 | <0.0001 |

two groups with different starting dates, with differences between groups evident during both post-treatment period 1 (Permanova pseudo-$F_{1,13}$: 2.96, $p = 0.001$) and post-treatment period 2 (Permanova pseudo-$F_{1,13}$: 2.01, $p = 0.001$), but not prior to treatment (Permanova pseudo-$F_{1,13}$: 1.05, $p = 0.47$; see electronic supplementary material, figure S8). These patterns all support the interpretation that antibiotic treatment does not in itself change the rumen microbiome, but that apparent differences between the treatment and control group were due to chance events during the initial randomization of cows into treatments.

### 3.5.2. Dung microbial communities

Mirroring the individual differences seen in the rumen microbiome, dung from individual cows differed in their microbiome composition (Permanova pseudo-$F_{7,38}$: 1.13, $p = 0.001$). However, contrasting with our findings for the rumen microbial communities, the dung microbiome was significantly impacted by the penicillin treatment in experiment 1 (figure 6). While treatment and control cattle microbiomes did not differ prior to antibiotic treatment (figure 6; Permanova pseudo-$F_{1,20}$: 1.03, $p = 0.36$), differences were evident two weeks after treatment (figure 6; Permanova pseudo-$F_{1,20}$: 1.73, $p = 0.003$). We also found a significant interaction between treatment and time (Permanova pseudo-$F$: 1.06, $p = 0.001$). These differences in the recovery rate of microbial communities from different antibiotic treatments were consistent across cow groups, despite some initial differences in the microbiomes among the two groups initiated at different dates (Permanova pseudo-$F_{1,20}$: 1.57, $p = 0.01$; for an illustration of patterns resolved by group, see electronic supplementary material, figure S9).

## 4. Discussion

Contrary to our first prediction, the antibiotics had no detectable effect on total gas emissions from cattle, effects on enteric fermentation were no larger than effects on gas emissions from manure, and antibiotics

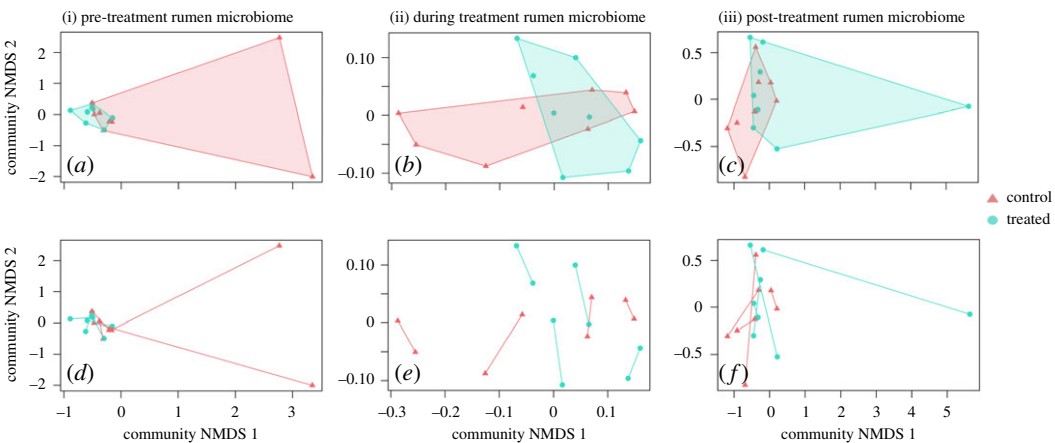

**Figure 5.** Effects of penicillin (experiment 1, Finland) on the microbiome composition of the rumen (i) before, (ii) during and (iii) after antibiotic treatment. These non-metric multidimensional scaling ordinations visually represent weighted UniFrac dissimilarities among samples in two dimensions. In the top panels, we separate samples by treatment, with cows in the control group identified in pink triangles and cows in the treatment group identified in turquoise circles. In the bottom panels, we separate samples by cow individual, with connector lines uniting samples from the same individual, and point shape and colour still corresponding to antibiotic treatment. Note that cow microbiomes tend to cluster by cow individual, rather than by antibiotic or control treatments, regardless of sampling period.

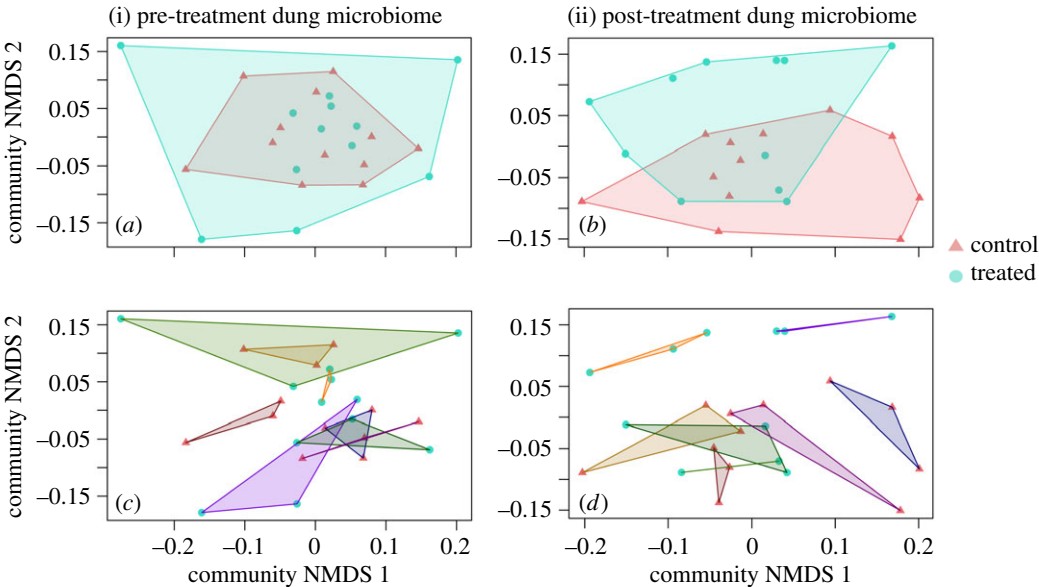

**Figure 6.** Effects of penicillin (experiment 1, Finland) on the microbiome composition of cattle dung (i) one week prior to antibiotic treatment and (ii) two weeks after antibiotic treatment. These non-metric multidimensional scaling ordinations visually represent weighted UniFrac dissimilarities among samples in two dimensions. In the top panels (*a,b*), we separate samples by treatment, with cows in the control group identified in pink triangles and cows in the treatment group identified in turquoise circles. In the bottom panels (*c,d*), polygons refer to individual cows with coloured connector lines uniting samples from the same individual, and point shape and colour still referring to antibiotic treatment. Note that cow microbiomes tend to cluster by cow individual, rather than by antibiotic or control treatments, regardless of sampling period. Dung microbial communities differ by cow identity prior to (Permanova pseudo-$F_{7,15}$: 1.64, $p = 0.001$) and two weeks after antibiotic treatment (Permanova pseudo-$F_{7,14}$: 1.90, $p = 0.001$).

actually decreased gas production *in vitro*. Consistent with our second prediction, microbiome-level effects of penicillin proved markedly different from those previously recorded for tetracycline. Contrary to our third prediction, effects of penicillin appeared similar regardless of the specific mode of administration. And consistent with our fourth prediction, effects on gas production differed substantially between dung maintained under aerobic versus anaerobic conditions. Overall, we attribute

these discrepancies between expectations and observations to the specificity of mechanisms triggered by individual antibiotic compounds, and to differences in modes of antibiotic administration. Below, we will outline this reasoning in full, while stressing that many, more specific studies will be needed before being able to summarize the net effect of antibiotics on livestock-derived greenhouse gas emissions.

## 4.1. Antibiotic effects vary with the compound used

Antibiotics vary in their composition of active compounds, which have diverse effects on specific microbial taxa. Overall, antibiotics can change the normal microbial content by inhibiting both pathogens and other, either harmless or beneficial bacteria, making it hard to predict the net effect at the level of the microbial community—and thereby the net effects in terms of gas emissions. In the current study, we focused on two antibiotics commonly used to treat cattle in Sweden [38,39] and elsewhere (e.g. [40]): penicillin and tetracycline. Of these, oxytetracycline is known as a broad spectrum antibiotic active against aerobic and anaerobic gram-positive and gram-negative bacteria, as well as against Mycoplasma and Chlamydia. Benzylpenicillin is active against gram-positive aerobic and anaerobic bacteria and usually against certain gram-negative bacteria within the genera *Pasteurella*, *Fusobacterium* and *Haemophilus* [41–43]. Importantly, these different microbes have different roles in $CH_4$ production, with bacteria releasing the hydrogen forming the substrate for $CH_4$ formation, but archaea being the actual methanogens (see [9]). In our previous study [21], we suggested that the use of tetracycline will shift the competitive balance between archaea and bacteria. Since archaea may be resistant to many classes of antibiotics [44], they could gain a competitive edge with antibiotic treatment. And since archaea are active methanogens, this may then translate to increased $CH_4$ emissions.

In the current study, microbiome-level effects of penicillin proved markedly different from those previously recorded for tetracycline [21]. While penicillin was administered intramuscularly in two independent experiments (experiments 1 and 2) and in the udder in a third treatment (experiment 2), we found no response in terms of gas emissions from either enteric fermentation or from dung exposed to pasture conditions. Since this lack of effect was matched by a lack of effect on the prokaryotic community of the rumen, and a matching lack of effect on feed intake, rumen fermentation and digestibility, one might conclude that the antibiotic never reached the digestive tract. Yet, such an interpretation is refuted by two facts: that we did observe a change in the microbiome composition of dung samples in experiment 1 (in response to penicillin), and a pronounced decrease in gas emissions from faeces, *in vitro* (in experiment 2, in response to both penicillin and tetracycline). The latter two findings offer clear evidence that both compounds will reach the digestive tract (see below). Thus, the overall effect of penicillin on the microbial community is simply different from that triggered by tetracycline (cf. [21]). The contrasting patterns found for tetracycline by Hammer *et al.* [21] may then be attributed to an insensitivity of methanogens to penicillin, as probably explained by the fact that their cell wall differs from that of bacteria [45]. Tetracycline, in contrast to penicillin, exerts its effect by inhibiting protein synthesis through preventing attachment of tRNA to the small subunit of the ribosome [46]. For methanogens, inhibition of protein synthesis by tetracycline is poorly studied. Hence, further research on the molecular mechanisms underlying antibiotic effects on rumen microbiome is clearly warranted to clarify the exact causal relationships—but what our study demonstrates is that the net effect of antibiotic compounds on greenhouse gas emissions differs substantially among antibiotic compounds.

## 4.2. Mode of administration did not affect response to antibiotics

In our experiments, we administered antibiotics by standard veterinary techniques, i.e. through intramuscular injection or locally in the udder. Such local or systemic administration of antibiotics might potentially be insufficient to prompt any effects on the digestive tract (but see above). As a contrast, we note that when used for growth promotion, antibiotics in non-therapeutic concentrations are added to the feed, potentially accentuating the effects on the rumen microbiome. Nevertheless in a study on antibiotics administered orally for growth promotion in beef cattle, no difference was found in the composition of the predominant intestinal bacteria [47], indicating that rumen and gastrointestinal microbiota may in some cases be resilient to direct exposure to antibiotics.

In our study, the concentration of the active component of the antibiotics might have been too low to prompt any effects in the rumen. Overall, we used doses approved and recommended by the Swedish and Finnish Medical Agency for treatment of infectious diseases. Hence, in order to trigger responses in the rumen, higher doses or oral administration may be needed. That antibiotic compounds did not

detectably affect the prokaryotic community composition or digestive processes of the rumen was supported by our metrics of community dissimilarity, rumen fermentation and digestibility.

However, the current doses did affect the microbes in the hindgut, since the composition of the microbiome markedly changed after administration of penicillin, and total gas production from dung *in vitro* was reduced in all antibiotic treatments compared to control. Since a close relationship between gas produced and substrate digested has been previously described by [48], and the gas production in faeces has been proposed as a good indicator of levels of hindgut fermentation [28], the lack of a match between gas production and digestive metrics seems surprising at first. Yet, two observations may account for the discrepancy: First, our observations of digestibility were derived from samples taken during post-treatment period 2, and were thus separated from the actual antibiotic treatment by two full weeks (see electronic supplementary material, figure S1). Second, in the current study, the largest inhibition of faecal gas production was observed for the broad-spectrum antibiotic oxytetracycline as applied in experiment 2, whereas our metrics of digestibility were based on the penicillin treatment applied in experiment 1. Both considerations will weaken the link between gas production and digestive metrics *a priori*, and rather emphasize the measurement of gas production as a sensitive probe for relevant effects on functionally important microbes.

Since the gas produced in the current bioassay consists of both $CO_2$ and $CH_4$ in unknown proportions with trace amounts of other gases [27,28], our current data are too limited to draw any firm conclusions about the specific effects of tetracycline on hindgut methanogens or methanogenesis. Yet, a few previous studies have investigated the effect of biogas production from manure from cows and pigs which have been treated with oxytetracycline [49,50]. Arikan *et al.* [51,52] used manure from calves, and found 27% lower cumulative biogas production from calves treated with oxytetracycline as compared to untreated calves. Thus, treatment with tetracycline will inhibit fermentation and is probably reflected in the digestion of feed. Further analysis on the causal relationships between fermentation patterns, digestibility and microbial community composition will hence be important to complement the current results. Importantly, what the patterns show in the current context is that each antibiotic compound has indeed reached the digestive tract regardless of the mode of administration, and that the effects of penicillin are similar regardless of the mode of administration (figure 4). The net effect of antibiotic compounds may then vary more with the active compounds (see previous section) than with the exact mode of administration.

## 4.3. Antibiotic effects vary with the response examined

In analysing the effects of antibiotics on gas emissions from cattle and cattle dung, effects on $CH_4$ emissions may be seen as particularly interesting, whereas emissions of $CO_2$ may actually offer a secondary concern—as should all carbon taken up as $CO_2$ by plants later be released in the same form (i.e. as $CO_2$) from dung, then cattle farming might actually be considered to be carbon neutral. Therefore, the main anthropogenic effect is the conversion of some of this carbon to the much more potent greenhouse gas of $CH_4$, and the concurrent emission of $N_2O$ [22].

In the current study, we found no effects of any of the antibiotic compounds used on either $CH_4$ emissions from enteric fermentation or on $CO_2$ emissions, as reflecting both feed digestion and overall metabolism. This was a major surprise, since we had expected the effects on gas emissions from enteric fermentation to vastly exceed those previously observed from dung pats [21]. Yet, for a different response—gas emissions under anaerobic conditions—all antibiotic compounds caused a marked decrease in emissions. Here, the effect size was large, with tetracycline causing a full 46% reduction in cumulative gas production compared to the untreated control.

The current results differ markedly from those of Hammer *et al.* [21], who examined effects on dung pats maintained under field conditions. In such drying-out pats, conditions get increasingly aerobic, and as a likely result, we tend to see a clear peak in $CH_4$ emissions some days after deposition/defecation followed by a drastic decline (see [22]). In a similar vein, aeration of the pats by dung beetle tunnels tend to decrease $CH_4$ emissions further [22]. In the current experiment, we recorded gas production under anaerobic conditions in closed syringes, under conditions akin to those of a biogas reactor. Here, conditions similar to those of a fresh dung pat will prevail over time, and will thus be triggering and/or sustaining processes different from those in a real dung pat. What is more, the temporal pattern of an initial increase in gas production followed by declining rates may be attributed to a decline in the amount of the most easily metabolized substrates available to microbes. Overall, this reveals the extent to which antibiotic effects may vary depending on the mechanisms triggered, and the response examined.

## 4.4. Everything varies with the cow examined

For almost every metric examined, potential effects of antibiotics were tested against a background of strong inter-individual variation. Whether we examined milk production, microbiome composition or digestibility, cow-to-cow variation was large *before* the treatment, and cows characterized by initially high values typically remained so throughout the experiment and vice versa (see baseline effects in electronic supplementary material, tables S2–S6). This variation emerged despite the fact that we had deliberately selected animals at a similar stage of lactation, of equal parity, antibiotic history (or rather the lack of it) etc. Individual variation is indeed one of the main patterns emerging from this study— and this pattern is consistent with that found in several previous studies [8,21,53]. Taken at face value, this variation may then cause one to ask whether relevant effects of antibiotics were not masked by small sample size coupled with large variation. In this context, we stress the crucial distinction between statistical power and biological relevance. The experimental animals were explicitly selected to be similar in as many respects as possible. If variation among these sets is still very much larger than variation due to antibiotic treatment, then there is only one sound conclusion: that the effect of antibiotics is not biologically relevant. In other words, if even here, in our carefully designed experiment, one cannot detect an effect, then it will seem that in the more motley cattle herds typical of any real barn, antibiotic treatment will not matter as compared to other sources of variation— whereas where we do find significant effects, these effects are strong enough to make a difference against the background of individual variation characteristic of any real herd. Increased sample size would clearly improve precision and power, but would hardly change the biological inference. These are essential considerations in gauging our results.

## 5. Conclusion

Taken together, our findings demonstrate compound- and context-dependent impacts of antibiotics on $CH_4$ emissions and underlying processes, against a background of strong inter-individual variation in almost every metric examined. Most importantly, our extensive exploration reveals multiple and partly opposing effects. Improved monitoring and estimates of agricultural antibiotic use will thus be necessary to identify how much antibiotics may impact the overall contribution of livestock production to global warming. Such insights are urgently needed—since wherever we look, some of the relative effect sizes consistently prove large (compare [21] versus figure 4, current study) whereas the absolute emissions affected are immense ([16], and references therein). What complicates our search for global estimates of antibiotic effects is the large inter-individual variation characteristic of cattle. Thus, in searching for biologically relevant patterns, and in weighing variation attributed to antibiotics against effects caused by other factors, we should remain careful in selecting samples representative of the intended reference population. As an important caveat, we do not know to what degree our current findings apply beyond our target population of Nordic dairy cattle. While our work indicates that antibiotic administration can disrupt microbiomes and the production of certain gases, more work is required before drawing general conclusions about antibiotic effects on global cattle production.

Animal ethics. All procedures used conform to the legal requirements of the countries in which the work was carried out (Finland and Sweden) and to all institutional guidelines (Swedish University of Agricultural Sciences, SLU, and Natural Resources Institute Finland, Luke). All handling of animals in experiment 1 (Finland) was approved by the Regional State Administrative Agency permissions (ESAVI/3207/04.10.07/2017, Hämeenlinna, Finland) in accordance with the guidelines established by the European Community Council Directives 497 and 564/2013/EEC (European Union, 2013). All handling of animals in experiment 2 (Sweden) was approved by the Umeå Ethics Committee for Animal Research, Sweden (Dnr. A 29-2017).

Permission to carry out fieldwork. Permission for fieldwork at the Viikki experimental farm was granted by its director, Dr Miika Kahelin.

Data accessibility. The datasets supporting this article are available at the Dryad Digital Repository: https://doi.org/10.5061/dryad.t7f3rc8 [54].

Authors' contributions. All authors contributed to conceive and design the work. For experiment 1 (Finland), I.T. and A.-R.B. implemented all parts of the experiment involving the live cows, whereas J.L. performed the experiment on gas emissions from pats under field conditions and performed the laboratory and statistical analyses of prokaryotic community composition, with added input from T.H. R.D., M.R., J.D. and S.A. were responsible for the full implementation of experiment 2 (Sweden). A.-R.B., T.R. and R.D. conducted the statistical analyses of all other

responses. T.R. and R.D. wrote a first draft of the manuscript, on which all authors provided substantial input. All authors gave final approval for publication.

Competing interests. The authors declare no competing interests.

Funding. Funding from the Swedish Cultural Foundation in Finland, through a stipend to R.D., is gratefully acknowledged. The Swedish part of the study was funded by the Swedish Research Council, FORMAS. Finnish experiment was partially supported by internal funding from the Natural Resources Institute Finland (Luke). Additional funding for this work (to J.L.) was provided by the National Science Foundation under grant no. DDIG 1701831 and a Graduate Research Opportunity Worldwide grant.

Acknowledgements. We gratefully acknowledge support by Bess Hardwick, University of Helsinki, in implementing key parts of the Finnish field experiment; Paula Lidauer from Luke for performing antibiotic treatment and to the Luke research barn staff for taking care of the animals and assistance in sample collection; Reija Danielsson and her crew at SLUs research barn in Umeå for taking care of the animals and assisting in sample collection.

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
