## [Reviewer comments · Royal Society Open Science]

Review History

RSOS-182049.R0 (Original submission)

Review form: Reviewer 1

Is the manuscript scientifically sound in its present form?

No

Are the interpretations and conclusions justified by the results?

No

Is the language acceptable?

Yes

Is it clear how to access all supporting data?

No

Do you have any ethical concerns with this paper?

No

Have you any concerns about statistical analyses in this paper?

No

Recommendation?

Major revision is needed (please make suggestions in comments)

Comments to the Author(s)

I consider the Authors have done a significant revision if comparing the current version against the prior one. However, there are some points deserving an additional efforts.

At their response, they claim [they] 'used separate sensors to record the head position of the cows'.

- I suggest Authors to properly describe these tools/sensors in terms of age, accuracy and previous number of uses.

Additionally, two points remain as improperly reflected:

- The " discussion about the concavity of the function related to Gas Production is missing, namely a discussion about the high increasing rates at the first weeks and the stabilized states afterwards"

- And the robustness to alternative methods of analyzing these samples.

Review form: Reviewer 2

Is the manuscript scientifically sound in its present form?

Yes

Are the interpretations and conclusions justified by the results?

Yes

Is the language acceptable?

Yes

Is it clear how to access all supporting data?

Yes

Do you have any ethical concerns with this paper?

No

Have you any concerns about statistical analyses in this paper?

No

Recommendation?

Accept with minor revision (please list in comments)

Comments to the Author(s)

The manuscript "Compound- and context-dependent effects of antibiotics on greenhouse gas emissions from livestock" is much improved from the earlier version submitted to Proceedings B. The authors took the reviewer's responses into account for the submission to Royal Society Open Science. I appreciate their effort and attention. There remain places where the limitations of the

study can be further clarified. These are indicated in specific comments below and need to be addressed prior to publication.

The manuscript also has multiple typographic errors, some of which are identified below.

Finally, more specific annotation of two figures is suggested that should make the results easier for a reader to quickly interpret.

Line 26 - "in" instead of "to farm animals"

Line 30 "...live microbial communities..."

Line 37 - extra period present

Line 77-80 "emissions from dairy cows in two in vivo experiments, referred to as Experiment 1 and Experiment 2. We used a randomized design to administer two different antibiotic compounds (benzylpenicillin or tetracycline) in two different ways (intramuscularly Intramammary) (Fig 1, left-hand side) and measured gas emissions from enteric fermentation in metabolic chambers or using GreenFeed system (Fig 1, mid-section)."

Thank you for trying to clarify your methods, however, Exp 1 only addressed control vs 1 antibiotic applied using 1 application method and metabolic chambers. Exp 2. In the remainder of this paragraph the differences between experiment 1 and 2 are more accurately reported. Line 77-80 seems to suggest that both experiments had multiple antibiotics and application methods.

Line 103 - change "measures" to "measured"

Line 165 - "...to get pelleted concentrates" - concentrates of what? Their standard diet?

Line 191 - "in duplicate" - however, Figures 6c and d show triplicate sampling of dung microbiome.

Line 309 - "Prokaryotic community composition. Antibiotic treatment had no detectable effect on the prokaryotic community composition of the rumen, but did impact the community composition of cow dung." - prokaryotic community composition was only analyzed using one antibiotic. These results need to be clarified as such.

Starting with Line 313 - similarly - ruman microbiome variation was only performed in Exp 1.

The way this is reported suggests that all antibiotics applied in different ways had the same effect. However, that was not explicitly tested.

Line 345 "Consistent with our second prediction, microbiome-level effects of penicillin proved markedly different from those previously recorded for tetracycline." This specific prediction seems like a "straw man" considering the vastly different mechanism of action that these two antibiotics employ.

Line 375 "Since this lack of effect was matched by a lack of effect on the prokaryotic community of the rumen, and a matching lack of effect on feed intake, rumen fermentation and digestibility, one might conclude that the antibiotic never reached the digestive tract. Yet, such an interpretation is refuted by two facts: that we did observe a change in the microbiome composition of dung samples, and a pronounced decrease in gas emissions from feces, in vitro."

- As I understand it, direct prokaryotic community results were measured in Exp 1 using only penicillin injected intramuscularly. Exp 2 using penicillin injected using 2 different methods and different antibiotics never directly measured prokaryotic community response in rumen or dung. Yes, there was a decrease in gas emissions, but whether this is due to changes in the microbiome composition of the dung sample does not seem to be directly measured. This difference in direct vs indirect evidence for prokaryotic community changes is stated more clearly in 414 and should be clarified more throughout the manuscript. It is surprising that given a sample size of only 24 animals for Exp 2, that dung microbiome samples were not taken and analyzed.

Figures

The panels in Figure 5 and 6 need to be more clearly annotated as to treatment condition - more similar to what was done in Figure S9 than the figures in the main text. Additionally, the colors used for individual cows can be difficult to discern (in particular the pink and purple colors). Individual cow colors should also correspond with control versus treatment. For example, all control are green and blue shades, while treatment are oranges and red. Alternatively, a different point shape for control vs treatment could also help.

Figure 6 – Conditions of panels A and B are included in the figure legend, but c and d are not. They can be inferred that a and c have similar conditions to b and d, but this should be explicitly stated.

Decision letter (RSOS-182049.R0)

11-Feb-2019

Dear Dr Danielsson,

The editors assigned to your paper ("Compound- and context-dependent effects of antibiotics on greenhouse gas emissions from livestock") have now received comments from reviewers. We would like you to revise your paper in accordance with the referee and Associate Editor suggestions which can be found below (not including confidential reports to the Editor). Please note this decision does not guarantee eventual acceptance.

Please submit a copy of your revised paper before 06-Mar-2019. Please note that the revision deadline will expire at 00.00am on this date. If we do not hear from you within this time then it will be assumed that the paper has been withdrawn. In exceptional circumstances, extensions may be possible if agreed with the Editorial Office in advance. We do not allow multiple rounds of revision so we urge you to make every effort to fully address all of the comments at this stage. If deemed necessary by the Editors, your manuscript will be sent back to one or more of the original reviewers for assessment. If the original reviewers are not available, we may invite new reviewers.

- Data accessibility

It is a condition of publication that all supporting data are made available either as supplementary information or preferably in a suitable permanent repository. The data accessibility section should state where the article's supporting data can be accessed. This section

should also include details, where possible of where to access other relevant research materials such as statistical tools, protocols, software etc can be accessed. If the data have been deposited in an external repository this section should list the database, accession number and link to the DOI for all data from the article that have been made publicly available. Data sets that have been deposited in an external repository and have a DOI should also be appropriately cited in the manuscript and included in the reference list.

If you wish to submit your supporting data or code to Dryad (<http://datadryad.org/>), or modify your current submission to dryad, please use the following link:
<http://datadryad.org/submit?journalID=RSOS&manu=RSOS-182049>

- **Competing interests**

- **Authors' contributions**

- **Acknowledgements**

- **Funding statement**

on behalf of Prof Kevin Padian (Subject Editor)
openscience@royalsociety.org

Associate Editor's comments:

Associate Editor: 1

Comments to the Author:

Your transferred manuscript has been assessed by a number of the original reviewers of the paper. They are broadly positive towards the manuscript, and your efforts to improve this. A number of outstanding items remain, and you must respond to these, and provide explanations in the point-by-point response you include with the revision, otherwise, this is a very promising revision -- well done!

Editor comments:

Thanks for your attention to previous comments. As you can see the reviewers still have some concerns. Please address these assiduously; as per our policy if the reviewers are not satisfied with the next revision we will be unable to consider the MS further. Good luck!

Comments to Author:

Reviewers' Comments to Author:

Reviewer: 1

Comments to the Author(s)

I consider the Authors have done a significant revision if comparing the current version against the prior one. However, there are some points deserving an additional efforts.

At their response, they claim [they] 'used separate sensors to record the head position of the cows'.

- I suggest Authors to properly describe these tools/sensors in terms of age, accuracy and previous number of uses.

Additionally, two points remain as improperly reflected:

- The " discussion about the concavity of the function related to Gas Production is missing, namely a discussion about the high increasing rates at the first weeks and the stabilized states afterwards"

- And the robustness to alternative methods of analyzing these samples.

Reviewer: 2

Comments to the Author(s)

The manuscript "Compound- and context-dependent effects of antibiotics on greenhouse gas emissions from livestock" is much improved from the earlier version submitted to Proceedings B. The authors took the reviewer's responses into account for the submission to Royal Society Open Science. I appreciate their effort and attention. There remain places where the limitations of the study can be further clarified. These are indicated in specific comments below and need to be addressed prior to publication.

The manuscript also has multiple typographic errors, some of which are identified below.

Finally, more specific annotation of two figures is suggested that should make the results easier for a reader to quickly interpret.

Line 26 - "in" instead of "to farm animals"

Line 30 "...live microbial communities..."

Line 37 - extra period present

Line 77-80 "emissions from dairy cows in two in vivo experiments, referred to as Experiment 1 and Experiment 2. We used a randomized design to administer two different antibiotic compounds (benzylpenicillin or tetracycline) in two different ways (intramuscularly

Intramammary) (Fig 1, left-hand side) and measured gas emissions from enteric fermentation in metabolic chambers or using GreenFeed system (Fig 1, mid-section)."

Thank you for trying to clarify your methods, however, Exp 1 only addressed control vs 1 antibiotic applied using 1 application method and metabolic chambers. Exp 2. In the remainder of this paragraph the differences between experiment 1 and 2 are more accurately reported. Line 77-80 seems to suggest that both experiments had multiple antibiotics and application methods.

Line 103 - change "measures" to "measured"

Line 165 - "...to get pelleted concentrates" - concentrates of what? Their standard diet?

Line 191 - "in duplicate" - however, Figures 6c and d show triplicate sampling of dung microbiome.

Line 309 - "Prokaryotic community composition. Antibiotic treatment had no detectable effect on the prokaryotic community composition of the rumen, but did impact the community composition of cow dung." - prokaryotic community composition was only analyzed using one antibiotic. These results need to be clarified as such.

Starting with Line 313 - similarly - ruman microbiome variation was only performed in Exp 1. The way this is reported suggests that all antibiotics applied in different ways had the same effect. However, that was not explicitly tested.

Line 345 "Consistent with our second prediction, microbiome-level effects of penicillin proved markedly different from those previously recorded for tetracycline." This specific prediction seems like a "straw man" considering the vastly different mechanism of action that these two antibiotics employ.

Line 375 "Since this lack of effect was matched by a lack of effect on the prokaryotic community of the rumen, and a matching lack of effect on feed intake, rumen fermentation and digestibility, one might conclude that the antibiotic never reached the digestive tract. Yet, such an interpretation is refuted by two facts: that we did observe a change in the microbiome composition of dung samples, and a pronounced decrease in gas emissions from feces, in vitro."

- As I understand it, direct prokaryotic community results were measured in Exp 1 using only penicillin injected intramuscularly. Exp 2 using penicillin injected using 2 different methods and different antibiotics never directly measured prokaryotic community response in rumen or dung. Yes, there was a decrease in gas emissions, but whether this is due to changes in the microbiome composition of the dung sample does not seem to be directly measured. This difference in direct vs indirect evidence for prokaryotic community changes is stated more clearly in 414 and should be clarified more throughout the manuscript. It is surprising that given a sample size of only 24 animals for Exp 2, that dung microbiome samples were not taken and analyzed.

Figures

The panels in Figure 5 and 6 need to be more clearly annotated as to treatment condition - more similar to what was done in Figure S9 than the figures in the main text. Additionally, the colors used for individual cows can be difficult to discern (in particular the pink and purple colors). Individual cow colors should also correspond with control versus treatment. For example, all control are green and blue shades, while treatment are oranges and red. Alternatively, a different point shape for control vs treatment could also help.

Figure 6 - Conditions of panels A and B are included in the figure legend, but c and d are not. They can be inferred that a and c have similar conditions to b and d, but this should be explicitly stated.

Author's Response to Decision Letter for (RSOS-182049.R0)

See Appendix A.

RSOS-182049.R1 (Revision)

Review form: Reviewer 1

Is the manuscript scientifically sound in its present form?

No

Are the interpretations and conclusions justified by the results?

No

Is the language acceptable?

Yes

Is it clear how to access all supporting data?

No

Do you have any ethical concerns with this paper?

No

Have you any concerns about statistical analyses in this paper?

Yes

Recommendation?

Reject

Comments to the Author(s)

I did not find sufficient the explanations nor the changes in the revised manuscript.

Review form: Reviewer 3

Is the manuscript scientifically sound in its present form?

No

Are the interpretations and conclusions justified by the results?

Yes

Is the language acceptable?

Yes

Is it clear how to access all supporting data?

Yes

Do you have any ethical concerns with this paper?

No

Have you any concerns about statistical analyses in this paper?

No

Recommendation?

Major revision is needed (please make suggestions in comments)

Comments to the Author(s)

This paper presents the effect of antibiotics (penicillin and tetracycline) on methane emissions and microbial composition (rumen and dung) assessed in two different experiments. Contrary to their hypotheses, authors showed that antibiotics did not increase the environmental impact of lactating dairy cows.

The manuscript is well written. Introduction, objectives and experimental design are clear, so I fully support the publication of these data. However, for reading easiness, I would prefer to split this paper in two separate manuscripts. This will allow authors to detail their Material and Methods (for instance methane measurement procedures, feeding regimen...) and Results sections. There is way too much supplementary materials, the reader always need to report to them, which complicate the understanding. Detailing the Material and Methods would also allow authors to delete Figure 1 and Table 1 (GreenFeed is missing anyway), which require a lot of energy to be understood! In Results, not having raw numerical values in Tables are very unusual. I strongly recommend authors to add them. It allows comparison with other papers.

Minor comments and questions:

- L139: What do you mean by blocking animals according to udder health ?
- L179-182: Which method did you use for digestibility measurement? Did you use different ones according to the period?
- L236: In statistical analyses, you have an "experiment 1" section but not an "experiment 2".
- L289: Throughout results, table numbers do not fit. For instance, here you mention Table 1 instead of Table 2.
- L355: What do you mean by "whole cattle"?

Decision letter (RSOS-182049.R1)

09-Aug-2019

Dear Dr Danielsson,

Manuscript ID RSOS-182049.R1 entitled "Compound- and context-dependent effects of antibiotics on greenhouse gas emissions from livestock" which you submitted to Royal Society Open Science, has been reviewed. The comments of the reviewer(s) are included at the bottom of this letter.

Please submit a copy of your revised paper before 01-Sep-2019. Please note that the revision deadline will expire at 00.00am on this date. If we do not hear from you within this time then it will be assumed that the paper has been withdrawn. In exceptional circumstances, extensions may be possible if agreed with the Editorial Office in advance. We do not allow multiple rounds of revision so we urge you to make every effort to fully address all of the comments at this stage. If deemed necessary by the Editors, your manuscript will be sent back to one or more of the original reviewers for assessment. If the original reviewers are not available we may invite new reviewers.

To revise your manuscript, log into <http://mc.manuscriptcentral.com/rsos> and enter your Author Centre, where you will find your manuscript title listed under "Manuscripts with Decisions." Under "Actions," click on "Create a Revision." Your manuscript number has been

appended to denote a revision. Revise your manuscript and upload a new version through your Author Centre.

- Ethics statement

- Data accessibility

- Competing interests

- Authors' contributions

- Acknowledgements

- Funding statement

on behalf of the Associate Editor, and Professor Kevin Padian (Subject Editor)
openscience@royalsociety.org

Associate Editor Comments to Author:

Thank you for your attention to the previous reviewers' comments. As you can see, the one who was originally highly critical of the manuscript is not completely satisfied. My sense is that you responded reasonably to their concerns, which seemed reasonable but were not severe enough to cause us to reject the manuscript. We have engaged a second reviewer who is overall positive about the manuscript and offers some useful comments that we hope you can address. The suggestion to make two papers would perhaps be more appropriate if this were a print journal with an online Supp Info section, but we have no restriction on length and so as long as your manuscript is clear and well organized with respect to its parts there is no problem with that. Best wishes for your revision.

Reviewer comments to Author:

Reviewer: 1
Comments to the Author(s)

I did not find sufficient the explanations nor the changes in the revised manuscript.

Reviewer: 3
Comments to the Author(s)

This paper presents the effect of antibiotics (penicillin and tetracycline) on methane emissions and microbial composition (rumen and dung) assessed in two different experiments. Contrary to their hypotheses, authors showed that antibiotics did not increase the environmental impact of lactating dairy cows.

The manuscript is well written. Introduction, objectives and experimental design are clear, so I fully support the publication of these data. However, for reading easiness, I would prefer to split this paper in two separate manuscripts. This will allow authors to detail their Material and Methods (for instance methane measurement procedures, feeding regimen...) and Results sections. There is way too much supplementary materials, the reader always need to report to them, which complicate the understanding. Detailing the Material and Methods would also allow

authors to delete Figure 1 and Table 1 (GreenFeed is missing anyway), which require a lot of energy to be understood! In Results, not having raw numerical values in Tables are very unusual. I strongly recommend authors to add them. It allows comparison with other papers.

Minor comments and questions:

- L139: What do you mean by blocking animals according to udder health ?
- L179-182: Which method did you use for digestibility measurement? Did you use different ones according to the period?
- L236: In statistical analyses, you have an “experiment 1” section but not an “experiment 2”.
- L289: Throughout results, table numbers do not fit. For instance, here you mention Table 1 instead of Table 2.
- L355: What do you mean by “whole cattle”?

Author's Response to Decision Letter for (RSOS-182049.R1)

See Appendix B.

RSOS-182049.R2 (Revision)

Review form: Reviewer 1

Is the manuscript scientifically sound in its present form?

No

Are the interpretations and conclusions justified by the results?

No

Is the language acceptable?

Yes

Do you have any ethical concerns with this paper?

No

Have you any concerns about statistical analyses in this paper?

Yes

Recommendation?

Reject

Comments to the Author(s)

Check previous comments.

Review form: Reviewer 3

Is the manuscript scientifically sound in its present form?

Yes

Are the interpretations and conclusions justified by the results?

Yes

Is the language acceptable?

Yes

Do you have any ethical concerns with this paper?

No

Have you any concerns about statistical analyses in this paper?

No

Recommendation?

Accept with minor revision (please list in comments)

Comments to the Author(s)

I have 3 last comments:

- L329 : You did not remove « experiment 1 » title as mentioned in your answer
- In Figure 2, the white color is not explained in the caption
- L393-304: The first part of the sentence needs to be rephrased (English wording)

Decision letter (RSOS-182049.R2)

16-Sep-2019

Dear Dr Danielsson:

On behalf of the Editors, I am pleased to inform you that your Manuscript RSOS-182049.R2 entitled "Compound- and context-dependent effects of antibiotics on greenhouse gas emissions from livestock" has been accepted for publication in Royal Society Open Science subject to minor revision in accordance with the referee suggestions. Please find the referees' comments at the end of this email.

The reviewers and Subject Editor have recommended publication, but also suggest some minor revisions to your manuscript. Therefore, I invite you to respond to the comments and revise your manuscript.

- **Ethics statement**

- Data accessibility

<http://datadryad.org/submit?journalID=RSOS&manu=RSOS-182049.R2>

- Competing interests

- Authors' contributions

- Acknowledgements

- Funding statement

Because the schedule for publication is very tight, it is a condition of publication that you submit the revised version of your manuscript before 25-Sep-2019. Please note that the revision deadline will expire at 00.00am on this date. If you do not think you will be able to meet this date please let me know immediately.

Kind regards,
Lianne Parkhouse
Editorial Coordinator
Royal Society Open Science
openscience@royalsociety.org

on behalf of the Associate Editor and Kevin Padian (Subject Editor)
openscience@royalsociety.org

Associate Editor Comments to Author:

While one of the reviewers consulted continues to express their concerns about the manuscript,

the Editors have taken the view that the majority of the reviewers consulted have considered the efforts you've made to improve the paper to be sufficient to make it publishable. There remain a number of minor matters to attend to, but we consider it better for the paper to be in the public domain and to allow further open debate to take place if there are truly claims remaining to be satisfied in the work.

Reviewer comments to Author:

Reviewer: 3

I have 3 last comments:

- L329 : You did not remove « experiment 1 » title as mentioned in your answer
- In Figure 2, the white color is not explained in the caption
- L393-304: The first part of the sentence needs to be rephrased (English wording)

Reviewer: 1

Check previous comments.

Author's Response to Decision Letter for (RSOS-182049.R2)

See Appendix C.

Decision letter (RSOS-182049.R3)

02-Oct-2019

Dear Dr Danielsson,

I am pleased to inform you that your manuscript entitled "Compound- and context-dependent effects of antibiotics on greenhouse gas emissions from livestock" is now accepted for publication in Royal Society Open Science.

Royal Society Open Science operates under a continuous publication model (<http://bit.ly/cpFAQ>). Your article will be published straight into the next open issue and this

will be the final version of the paper. As such, it can be cited immediately by other researchers. As the issue version of your paper will be the only version to be published I would advise you to check your proofs thoroughly as changes cannot be made once the paper is published.

Best regards,

Lianne Parkhouse
Royal Society Open Science
openscience@royalsociety.org

on behalf of the Associate Editor, and Professor Kevin Padian (Subject Editor)
openscience@royalsociety.org

Appendix A

Dear Editor,

Please find enclosed our revised manuscript entitled "Compound- and context-dependent effects of antibiotics on greenhouse gas emissions from livestock". We have considered the valuable comments and suggestions provided by the Editor and the Referees and have now made our best effort to improve the manuscript further. To identify and to justify the changes implemented, we have responded below point-by-point to each individual comment. For clarity, we have included the original comments *in italics*. The revisions implemented in the new version of the manuscript are presented as tracked changes.

Best regards,
Rebecca Danielsson, on behalf of all Co-authors

Comments by the Editors

Associate Editor's comments:

Associate Editor: 1

Comments to the Author:

Your transferred manuscript has been assessed by a number of the original reviewers of the paper. They are broadly positive towards the manuscript, and your efforts to improve this. A number of outstanding items remain, and you must respond to these, and provide explanations in the point-by-point response you include with the revision, otherwise, this is a very promising revision -- well done!

Editor comments:

Thanks for your attention to previous comments. As you can see the reviewers still have some concerns. Please address these assiduously; as per our policy if the reviewers are not satisfied with the next revision we will be unable to consider the MS further. Good luck!

RESPONSE: We thank the Editors for their kind comments. As will be evident below, we have made every effort to address the outstanding items. How we have modified the manuscript will be evident from the point-by-point responses below.

Comments to Author:

Reviewers' Comments to Author:

Reviewer: 1

Comments to the Author(s)

I consider the Authors have done a significant revision if comparing the current version against the prior one. However, there are some points deserving an additional efforts.

At their response, they claim [they] 'used separate sensors to record the head position of the cows'.

-I suggest Authors to properly describe these tools/sensors in terms of age, accuracy and previous number of uses.

RESPONSE: We thank Reviewer 1 for challenging us to specify every aspect of the equipment used. In terms of the *age, accuracy and previous number of uses* of the CH₄ and CO₂ gas sensors, we note that we have had the GreenFeed system in operation in our dairy barn since 2012, and are thoroughly experienced with its sources of error and reliability. (In 2015, we upgraded

(purchased) the system to also measure the O₂ consumption in dairy cows for energy balance studies. Since all instruments were carefully recalibrated at this point, and since the measurements in the current study were all made with the upgraded equipment, this upgrade does not affect the current readings in any way.) Our experience of the reliability and repeatability of measurements are highly positive. We note that as a rule of thumb, the exact number of times that the sensors have been used is less of a concern, since the sensors work normally until they fully stop working in one go. This is because we have relied on the NDIR method, which uses a light at one end of a tube and a sensor at the other end, and a light filter which only lets through light at a wavelength specifically absorbed by CH₄. The light source does eventually burn out, but that is easily noticed since then the sensor instantaneously stops working. We have naturally performed the normal calibrations (for concentration and recovery), which will adjust for potential drift over time. The sensor monitoring the position of the head of the cow is also robust and accurate, whereas the number of uses would be hard to specify. These further aspects have all been specified in the revised text (see Supplementary Online Material, section Quantification of gas emissions in vivo).

Additionally, two points remain as improperly reflected:

-The "discussion about the concavity of the function related to Gas Production is missing, namely a discussion about the high increasing rates at the first weeks and the stabilized states afterwards"

RESPONSE: In his/her report on the previous version of the manuscript, the Reviewer posed the same question like this:

Finally, a proper discussion about the concavity of the function related to Gas Production is missing, namely a discussion about the high increasing rates at the first weeks and the stabilized states afterwards. Therefore, a major point rises - can the use of antibiotics significantly reduce the initial values of gas production? If so, which are the global advantages and disadvantages for ecological systems, for industry and for society?

We directly responded that we felt he/she may have got it a bit wrong, since we were seriously worried about the risk for over interpretation. We noted that the metrics of gas production derived from limited amounts of dung substrate kept in airtight syringes. The initial increase in gas production followed by declining rates can then be attributed to a mere decline in the amount of the most easily-metabolized substrates available to microbes. Given that the two other reviewers had cautioned against major generalizations from restricted data, we explained that we had opted for not including new inferences regarding the global advantages and disadvantages for ecological systems, for industry and for society. Now we are unsure of whether the Reviewer understood our concern, since he/she simply repeats his/her original question with no reference to our prior response. To be as clear as possible about our position, we have now added a new section to the discussion where we discuss the reasons for the decelerating gas production, without making the suggested connections to global patterns, which we find unwarranted, this can be found in tracked changes in the discussion.

- And the robustness to alternative methods of analyzing these samples.

RESPONSE: Here, we are unsure of what the Reviewer is asking. Should the Editor agree that this is an outstanding item, then we must insist on an editorial steer over consultation with the Reviewer. There are two alternative interpretations:

- 1) The Reviewer may refer to the following question advanced during the previous round: "are these results robust to alternative methods like meta-analysis or additional controls?" If so, then we are at utter loss with respect to what we are expected to comment upon. Clearly, we do not hold additional controls, so referring to those would be mere speculation. And while we are thoroughly experienced with meta-analysis (e.g. Lewinsohn & Roslin. *Ecology Letters* 11: 398-416; Gripenberg [...] Roslin. *Ecology Letters* 13:383-393, both highly cited), we simply fail to see how it might fruitfully be applied here.

- 2) The Reviewer may refer to his/her previous question “*if Authors use bayesian methods, will they achieve convergent results?*”. If this is the case, then we are equally at loss with respect to the objective of the Reviewer’s recommendation. We reiterate that applying both frequentist and Bayesian methods to the same data set to simply verify the convergence of results (when there is no reason to expect anything else) would seem like a non-standard request. If the Reviewer has some strong prior in mind, then we would have needed his/her guidance to accommodate it in the analysis, along with a precise, well-reasoned alternative model specification. We explicitly asked for them in our previous response but only got the question repeated to ourselves.

Reviewer: 2

Comments to the Author(s)

The manuscript "Compound- and context-dependent effects of antibiotics on greenhouse gas emissions from livestock" is much improved from the earlier version submitted to Proceedings B. The authors took the reviewer’s responses into account for the submission to Royal Society Open Science. I appreciate their effort and attention. There remain places where the limitations of the study can be further clarified. These are indicated in specific comments below and need to be addressed prior to publication. The manuscript also has multiple typographic errors, some of which are identified below. Finally, more specific annotation of two figures is suggested that should make the results easier for a reader to quickly interpret.

RESPONSE: We thank the Reviewer for his/her careful reading and useful comments. We have done our best to improve the manuscript by further clarifying the limitations of our study. The text has been carefully checked by a native English-speaking person (TH). We have also done our best to clarify the figures and to improve their quality (see details in the responses below).

Line 26 - “in” instead of “to farm animals”

RESPONSE: changed as suggested

Line 30 “...live microbial communities...”

RESPONSE: changed as suggested

Line 37 – extra period present

RESPONSE: now deleted

Line 77-80 “emissions from dairy cows in two in vivo experiments, referred to as Experiment 1 and Experiment 2. We used a randomized design to administer two different antibiotic compounds (benzylpenicillin or tetracycline) in two different ways (intramuscularly Intramammary) (Fig 1, left-hand side) and measured gas emissions from enteric fermentation in metabolic chambers or using GreenFeed system (Fig 1, mid-section).”

Thank you for trying to clarify your methods, however, Exp 1 only addressed control vs 1 antibiotic applied using 1 application method and metabolic chambers. Exp 2. In the remainder of this paragraph the differences between experiment 1 and 2 are more accurately reported. Line 77-80 seems to suggest that both experiments had multiple antibiotics and application methods.

RESPONSE: We agree with Reviewer 2 that the section on ll. 77-80 was a bit vaguely phrased. We have now clarified the contents by stressing that the two experiments differed in the antibiotics used and treatments administered, as exactly illustrated in Fig. 1, see tracked changes in the introduction section.

Line 103 – change “measures” to “measured”

RESPONSE: changed as suggested

Line 165 – “...to get pelleted concentrates” – concentrates of what? Their standard diet?

RESPONSE: The contents of the pellets has now been detailed in new Table S1.2 in the Supplementary online material.

Line 191 – “in duplicate” – however, Figures 6c and d show triplicate sampling of dung microbiome.

RESPONSE: Both are actually correct, for the following reason: Fecal sample collection was performed in duplicate by placing fecal samples into 2 zip-lock bags. One bag was used for the microbiome analysis while the second bag is still kept stored as a reserve at -80°C. However, from each bag actually used, we then took three separate samples for DNA extraction, as shown in Figure 6. This has now been explained in the text.

Line 309 – “Prokaryotic community composition. Antibiotic treatment had no detectable effect on the prokaryotic community composition of the rumen, but did impact the community composition of cow dung.” – prokaryotic community composition was only analyzed using one antibiotic. These results need to be clarified as such.

RESPONSE: The Reviewer is certainly correct, and we have now clarified the text by writing “Antibiotic treatment with penicillin, in Experiment, had no detectable....” ,

Starting with Line 313 – similarly – ruman microbiome variation was only performed in Exp 1. The way this is reported suggests that all antibiotics applied in different ways had the same effect. However, that was not explicitly tested.

RESPONSE: Again, the Reviewer is certainly correct, and to clarify the intended meaning we have now changed the more general “antibiotic treatment” to “penicillin treatment”, and added “control or treated with penicillin” instead of discussing “difference between treatments”, see the highlighted parts in the “Prokaryotic community composition” paragraph. As a result, we hope that the whole section is now clear and specific enough.

Line 345 “Consistent with our second prediction, microbiome-level effects of penicillin proved markedly different from those previously recorded for tetracycline.” This specific prediction seems like a “straw man” considering the vastly different mechanism of action that these two antibiotics employ.

RESPONSE: There may perhaps be the shadow of a straw man falling over this prediction, yet again, we note that our previous results (Hammer et al. 2014 Proc B) have been widely publicized as “antibiotics resulting in [this and that]”. To point out that different compounds have widely different modes of actions *and thus* markedly different consequences may then be less self-evident than the Reviewer suggests. We note that *Royal Society Open Science* is a multidisciplinary journal reaching way beyond a readership trained in veterinary or microbial science, and that the paper should be written and read with this in mind.

Line 375 “Since this lack of effect was matched by a lack of effect on the prokaryotic community of the rumen, and a matching lack of effect on feed intake, rumen fermentation and digestibility, one might conclude that the antibiotic never reached the digestive tract. Yet, such an interpretation is refuted by two facts: that we did observe a change in the microbiome composition of dung samples, and a pronounced decrease in gas emissions from feces, in vitro.”

-As I understand it, direct prokaryotic community results were measured in Exp 1 using only penicillin injected intramuscularly. Exp 2 using penicillin injected using 2 different methods and different antibiotics never directly measured prokaryotic community response in rumen or dung. Yes, there was a decrease in gas emissions, but whether this is due to changes in the microbiome

composition of the dung sample does not seem to be directly measured. This difference in direct vs indirect evidence for prokaryotic community changes is stated more clearly in 414 and should be clarified more throughout the manuscript. It is surprising that given a sample size of only 24 animals for Exp 2, that dung microbiome samples were not taken and analyzed.

RESPONSE: Here, we must naturally agree that direct changes in prokaryotic community composition in rumen or dung were measured directly only in Experiment 1. Yet, we do feel that the Reviewer is adopting the attitude of the Devil's advocate to a more than necessary extent. We did find a direct effect of administering penicillin by one mode on the dung in Experiment 1, and we did find an effect on gas emissions in Experiment 2. That shows that the antibiotics entered the digestive tract. Our argument does not ultimately rely on establishing a causal relation between changes in gas emissions and changes in the gut microbiome. Exactly how the Reviewer suggests antibiotics may affect gas production from feces without entering the feces, when tens of metrics of digestion and digestibility show no change, is unclear. To ease the Reviewer's concerns, we have specified what direct evidence was gained from what experiment, both in the section here highlighted and elsewhere in the manuscript.

Figures

The panels in Figure 5 and 6 need to be more clearly annotated as to treatment condition – more similar to what was done in Figure S9 than the figures in the main text. Additionally, the colors used for individual cows can be difficult to discern (in particular the pink and purple colors). Individual cow colors should also correspond with control versus treatment. For example, all control are green and blue shades, while treatment are oranges and red. Alternatively, a different point shape for control vs treatment could also help.

Figure 6 – Conditions of panels A and B are included in the figure legend, but c and d are not. They can be inferred that a and c have similar conditions to b and d, but this should be explicitly stated.

RESPONSE: We thank the Reviewer for his/her very sharp-eyed and helpful suggestions for improving the Figures. Each of Fig. 5, Fig. 6, Fig. S8 and Fig. S9 (along with their legends) has now been revised for visual consistency, for internal coherence in contents, and for ease of interpretability. We believe that the new set is vastly better than the old one.

Appendix B

Dear Editor,

Please find enclosed our revised manuscript entitled "Compound- and context-dependent effects of antibiotics on greenhouse gas emissions from livestock". We are most grateful for your decision to reconsider our manuscript, and for consulting with a further reviewer – the valuable comments and suggestions of whom we have considered in full detail. These changes, we feel, have helped us further improve the manuscript. To identify and to justify the changes implemented, we have responded below point-by-point to each individual comment. For clarity, we have included the original comments *in italics*. The revisions implemented in the new version of the manuscript are presented as tracked changes. Given these changes, we hope and trust that the manuscript is now ready for publication in your excellent journal.

Best regards,
Rebecca Danielsson, on behalf of all co-authors

Comments by the Editors

Associate Editor's comments:

Associate Editor: 1

Comments to the Author:

Thank you for your attention to the previous reviewers' comments. As you can see, the one who was originally highly critical of the manuscript is not completely satisfied. My sense is that you responded reasonably to their concerns, which seemed reasonable but were not severe enough to cause us to reject the manuscript. We have engaged a second reviewer who is overall positive about the manuscript and offers some useful comments that we hope you can address. The suggestion to make two papers would perhaps be more appropriate if this were a print journal with an online Supp Info section, but we have no restriction on length and so as long as your manuscript is clear and well organized with respect to its parts there is no problem with that. Best wishes for your revision

RESPONSE: We thank you for reconsidering our manuscript, and have now made our best to further improve the manuscript guided by the comments and suggestions of Reviewer 3. As most of these concerned sections left unclear by our previous presentation, we have now done our utmost to clarify any opacities. All added information asked for has been added exactly as solicited.

With respect to the suggestion of splitting the manuscript in two, we have sided with the Editor's suggestion of keeping it all together, thereby making full use of the current lack of length restriction offered by the online format. In doing so, we have moved material from the Supplements to the main text, as suggested by the Reviewer. We note that the previous solution reflected the history of the manuscript (which was originally transferred to *Proceedings B*, with its strict length limits), and that the current reorganization of the text has clearly made the paper more readable. Thus, we would like to warmly thank the Reviewer for his/her constructive suggestion. How we have modified the manuscript will be evident from the point-by-point responses below.

Comments to Author:

Reviewers' Comments to Author:

Reviewer: 1

Comments to the Author(s)

I did not find sufficient the explanations nor the changes in the revised manuscript.

RESPONSE: Since the Reviewer's comment includes no specific pointers nor suggestions for further changes, we feel forced to refer to our previous responses. From the Associate Editor's comments, we understand that the Editor finds our previous responses reasonable.

Reviewer: 3

Comments to the Author(s)

This paper presents the effect of antibiotics (penicillin and tetracycline) on methane emissions and microbial composition (rumen and dung) assessed in two different experiments. Contrary to their hypotheses, authors showed that antibiotics did not increase the environmental impact of lactating dairy cows.

The manuscript is well written. Introduction, objectives and experimental design are clear, so I fully support the publication of these data. However, for reading easiness, I would prefer to split this paper in two separate manuscripts. This will allow authors to detail their Material and Methods (for instance methane measurement procedures, feeding regimen...) and Results sections. There is way too much supplementary materials, the reader always need to report to them, which complicate the understanding. Detailing the Material and Methods would also allow authors to delete Figure 1 and Table 1 (GreenFeed is missing anyway), which require a lot of energy to be understood! In Results, not having raw numerical values in Tables are very unusual. I strongly recommend authors to add them. It allows comparison with other papers.

RESPONSE: We thank the Reviewer for his/her careful reading and constructive comments. In response to the request for further readability, we have 1) sided with the Editor's suggestion of keeping it all together, given the lack of length restriction and the fact that all information is needed to answer the overarching questions addressed by our study; 2) carefully reworked the structure of the Materials and Methods section, as solicited by the Reviewer, moving material from the Supplements to the main text.

With respect to the request for "raw numerical values", we stress that all primary data are supplied in the Dryad data repository, and that the reader will face no problem in recreating each statistic reported in the paper. In contrast, adding raw data to all tables, or even adding least squares means per treatment, would a) double the size of the already-massive tables and thus add some ten pages to the manuscript, and b) create direct overlap with the figures showing the very same information. The values can be easily extracted using standard software free of charge, like ImageJ.

With respect for the request to *delete Figure 1 and Table 1 (GreenFeed is missing anyway), which require a lot of energy to be understood*, we have gone for a carefully-crafted compromise. First, we have carefully restructured the Materials and methods -text to include all information needed to make sense of the full material. At the same time, we have retained Fig. 1 and Table 1 as graphical summaries of what was done how (since we expect some readers to need this extra support). This dual solution will allow the reader to use the illustrations as a clarifying reference where needed, while circumventing the need for *a lot of energy to be understood*. As an added justification, the footnotes of Table 1 includes condensed information which will not possibly fit any realistically-sized materials and methods section – but will still be sought for by specific readers with specific interests. Having said that, we realized that the

GreenFeed system had accidentally gone missing from the table. It has now been added under Table section "Gas emissions from enteric fermentation".

Overall, we reiterate that the previous structure of the manuscript was a legacy of its previous submission history, and that the current reorganization of the text has made the paper much more readable than before. Thus, we remain much indebted to the Reviewer for his/her constructive suggestion.

Minor comments and questions:

- L139: *What do you mean by blocking animals according to udder health ?*

RESPONSE: We appreciate the reviewers comment and have rephrased the sentence to milk somatic cell count. Milk somatic cell count is used as a measurement of mammary gland inflammatory status and was taken in consideration in this experiment for two reasons. One reason was to ensure that the cows were healthy and did not suffer from a sub clinical mastitis, the other reason for blocking according to SCC was that this experiment was shared between two projects. Since all recruited cows in this experiment had a composite milk SCC below 150 000 cell/ml we are convinced that this would not affect methane emission, but would be relevant for the other project.

- L179-182: *Which method did you use for digestibility measurement? Did you use different ones according to the period?*

RESPONSE: This information was offered on

- Lines 179-182 as "Total-tract apparent digestibility coefficients were determined by using either indigestible neutral detergent fiber (NDF) as an internal marker (baseline period and post-treatment period 1) or total fecal collection over a 72-h interval starting at 1800 h on day 36 of post-treatment period 2 (Fig. S1)."
- Lines 188-191 as "Total-tract apparent digestibility of nutrients was then calculated as $100 - (\% \text{marker in feed} / \% \text{marker in excreta}) \times (\% \text{nutrient in excreta} / \% \text{nutrient in feed}) \times 100$ or $((\text{nutrient intake} - \text{nutrient excretion}) / \text{nutrient intake}) \times 100$ for marker and total collection methods, respectively."

In other words, we have used two different methods: for the baseline period and for Period 1, we used the internal marker of iNDF, whereas for Period 2, we used total collection of feces. This solution was based on the fact that while total collection of feces is arguably the best method to calculate nutrient digestibility, it is very laborious. Therefore, we used the less-laborious method of using indigestible NDF as an internal marker to measure digestibility during the baseline period and period 1. For the Period 2, we used the more-laborious method of measuring the total amount of excreted nutrients using total collection of feces, thereby maximizing our resolution in capturing any possible effects of treatments on digestibility. This has now been further clarified in text, see Line 211.

- L236: *In statistical analyses, you have an "experiment 1" section but not an "experiment 2".*

RESPONSE: The Reviewer is certainly correct, and we apologize for this mishap. In response, we have now removed the subtitle "Experiment 1" and instead clarified in text which study we refer to, explicitly referring to either Experiment 1 or 2 where the analysis differed between experiments.

- L289: *Throughout results, table numbers do not fit. For instance, here you mention Table 1 instead of Table 2.*

RESPONSE: Again, the Reviewer is absolutely right, and we have now double-checked all references to Table and Figure numbers throughout the text.

- L355: *What do you mean by "whole cattle"?*

RESPONSE: Here we refer to the total gas production from the whole animal, i.e. not only to emissions from e.g. burps or from faeces. To clarify the intended meaning, "whole cattle" has now been changed into ".. total gas emissions from cattle,"

Appendix C

Dear Editor,

Please find enclosed our revised manuscript entitled "Compound- and context-dependent effects of antibiotics on greenhouse gas emissions from livestock". We are very glad to hear that our manuscript has been accepted for publication in Royal Society Open Science, we have made the last changes according to the suggestions from reviewer 3. To identify and to justify the changes implemented, we have responded point-by-point to each individual comment. For clarity, we have included the original comments *in italics*. The revisions implemented in the new version of the manuscript are presented as tracked changes.

Best regards,
Rebecca Danielsson, on behalf of all co-authors

Reviewer: 3

We thank the reviewer for your thorough reading once more of the manuscript, we have made the last changes as suggested, see below in detail.

Comments to the Author(s)

L329 : You did not remove « experiment 1 » title as mentioned in your answer

RESPONSE: Thanks for paying attention to this, title "Experiment 1" has now been removed

In Figure 2, the white color is not explained in the caption

RESPONSE: The white color has now been explained in caption for figure 2

L393-304: The first part of the sentence needs to be rephrased (English wording)

RESPONSE: The sentence has now been rephrased, see Line 360-361 in new version